# Camrelizumab and apatinib plus induction chemotherapy and concurrent chemoradiotherapy in stage N3 nasopharyngeal carcinoma: a phase 2 clinical trial

Hu Liang[1,2,9], Yao-Fei Jiang[1,2,9], Guo-Ying Liu[1,3,9], Lin Wang[1,2,9], Jian-Wei Wang[2,4,9], Nian Lu[1,2,5], Wei-Xiong Xia[1,2], Liang-Ru Ke[2,4], Yan-Fang Ye[6], Jin-Lin Duan[2,7], Wei-Xin Bei[1,2], Shu-Hui Dong[1,2], Wang-Zhong Li[1,2], Li-Ting Liu[1,2], Chong Zhao[1,2] ✉, Changqing Xie [8] ✉ & Yan-Qun Xiang [1,2] ✉

The antiangiogenic agent apatinib has been shown to clinically improve responses to immune checkpoint inhibitors in several cancer types. Patients with N3 nasopharyngeal carcinoma have a high risk of distant metastasis, however, if the addition of immunotherapy to standard treatment could improve efficacy is unclear. In this phase II clinical trial (ChiCTR2000032317), 49 patients with stage $T_{any}N3M0$ nasopharyngeal carcinoma were enrolled and received the combination of three cycles of induction chemotherapy, camrelizumab and apatinib followed by chemoradiotherapy. Here we report on the primary outcome of distant metastasis-free survival and secondary end points of objective response rate, failure-free survival, locoregional recurrence-free survival, overall survival and toxicity profile. After induction therapy, all patients had objective response, including 13 patients (26.5%) with complete response. After a median follow-up of 28.7 months, the primary endpoint of 1-year distant metastasis-free survival was met for the cohort (1-year DMFS rate: 98%). Grade≥3 toxicity appeared in 32 (65.3%) patients, with the most common being mucositis (14[28.6%]) and nausea/vomiting (9[18.4%]). In this work, camrelizumab and apatinib in combination with induction chemotherapy show promising distant metastasis control with acceptable safety profile in patients with stage $T_{any}N3M0$ nasopharyngeal carcinoma.

Nasopharyngeal carcinoma (NPC) differs from other head and neck cancers with unique geographical, etiological, and biological features[1]. It is endemic in the area of Southern China, Southeast Asia, and North Africa. More than 70% of NPC patients are classified as having locoregionally advanced disease among the 129,000 newly diagnosed cases annually[1]. Induction chemotherapy (IC) followed by concurrent chemoradiotherapy (CCRT) is the current standard-of-care treatment option for locoregionally advanced NPC[2]. The application of intensity-modulated radiotherapy (IMRT) and combined chemoradiotherapy has substantially improved locoregional control.

However, distant metastasis predominates due to treatment failure in patients with locoregionally advanced NPC, especially for patients with N3 disease[1,3,4]. After radical chemoradiotherapy (CRT), 35%–50% patients with N3 disease develop distant metastasis, about half of which occur within 1 year[3–6]. Although treated with IC and CCRT, distant metastasis-free survival (DMFS) in patients with N3 disease still remains unsatisfactory, as reported by several phase III clinical trials[6–9], even with modification of IC regimen[9] or adjuvant metronomic capecitabine[10]. Therefore, novel treatment strategies are needed for patients with N3 disease to eradicate micrometastatic lesions and improve treatment outcomes, while maintaining acceptable safety profile.

The efficacy of immune checkpoint blockade has been confirmed in recurrent or metastatic (R/M) NPC in the first-, second-, and later-line settings[11,12], which is likely associated with intrinsic properties of NPC, including the abundant lymphocytic infiltrations and upregulation of programmed cell death-ligand 1 (PD-L1)[13–16]. Recently, immunotherapy in combination with standard treatment regimen in the curative setting was reported, however, with no benefit seen in N3 patients[17].

Camrelizumab is a humanized immunoglobulin G4/k programmed cell death-1 (PD-1) monoclonal antibody, which is well tolerated with antitumor activity in R/M NPC[15,16]. Meanwhile, antiangiogenic agent apatinib, an orally administered vascular endothelial growth factor receptor 2 (VEGFR2) tyrosine kinase inhibitor, has shown synergetic antitumoral efficacy in combination with camrelizumab in several solid tumors, including advanced NPC[18], hepatocellular carcinoma[19], breast cancer[20,21], and osteosarcoma[22]. In this work, we evaluated the efficacy and safety of the combination of camrelizumab and apatinib with IC and CCRT as a curative approach in stage TanyN3M0 NPC with the goal of significantly decreasing risk of distant metastasis.

## Results

### Patients and treatment

From May 2020 to July 2021, a total of 50 patients were enrolled. One patient whose diagnosis did not meet eligibility criteria was excluded from the study (Fig. 1). Table 1 lists baseline characteristics of the remaining 49 patients (intention-to-treat population). Among these 49 patients, one patient refused CCRT after induction therapy and one

**Table 1 | Characteristics of the patients at baseline**

| Characteristic | No. (%) of patients (N = 49) |
|---|---|
| Age, year | |
| Median (interquartile range) | 46 (37–52) |
| Range | 20–65 |
| Gender | |
| Male | 37 (75.5) |
| Female | 12 (24.5) |
| ECOG perform status[a] | |
| 0 | 39 (79.6) |
| 1 | 10 (20.4) |
| Tumor category[b] | |
| T1 | 3 (6.1) |
| T2 | 7 (14.3) |
| T3 | 26 (53.1) |
| T4 | 13 (26.5) |
| Lymph node levels | |
| Retropharyngeal | 48 (98.0) |
| I | 9 (18.4) |
| II | 49 (100.0) |
| III | 49 (100.0) |
| IV | 40 (81.6) |
| IVA | 28 (57.1) |
| IVB | 18 (36.7) |
| >6 cm | 13 (26.5) |
| Histology classification | |
| Non-keratinising differentiated | 1 (2.0) |
| Non-keratinising undifferentiated | 48 (98.0) |
| Plasma EBV DNA level, median (interquartile range), copies/mL[c] | 4710 (1670–17900) |

[a]Eastern Cooperative Oncology Group (ECOG) performance-status scores range from 0 to 5, with higher scores indicating greater disability.
[b]Tumor category refer to the T stage according to the TNM staging system (American Joint Committee on Cancer, 8th edition).
[c]Shown are the data of pretreatment plasma Epstein-Barr virus (EBV) DNA level which were based on a lower limit of detection cutoff value of 1000 copies/mL.

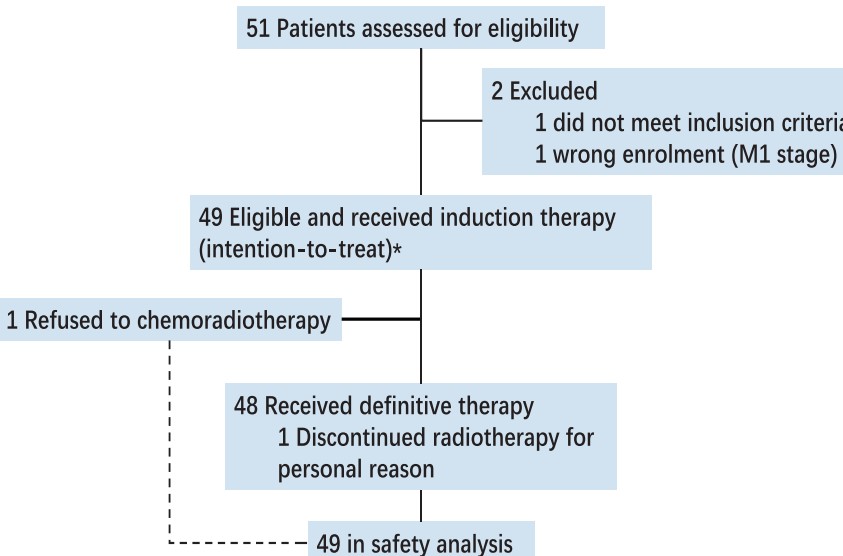

**Fig. 1 | Trial profiles.** The induction therapy included NAB-paclitaxel, cisplatin and capecitabin regimen induction chemotherapy, antiangiogenic agent apatinib, and anti-PD-1 immune checkpoint inhibitor antibody camrelizumab. *With the exception of the primary end point (based on the first 46 patients per protocol).

patient discontinued radiotherapy after receiving 9 fractions due to personal reasons.

## Compliance

All the eligible 49 patients started protocol-defined induction therapy and were included in the safety analysis (Fig. 1). All 49 patients (100%) completed 3 cycles of IC. Five patients (10.2%) had dose reductions of nab-paclitaxel, cisplatin, or capecitabine, mainly due to non-hematological toxic effects (4 of 5 patients; Supplementary Table 5).

Of 49 patients, 44 (89.8%) received at least one cycle of apatinib, whereas five patients discontinued apatinib early due to rash (1/5), exfoliative dermatitis (1/5), nasal bleeding (1/5), treatment-unrelated oral wound (1/5), and patient refusal (1/5). Twenty-six patients (53.1%) required one or more dose interruptions of apatinib. Three patients (6.1%) required reduction of apatinib dose. Overall, the median dose intensity of apatinib was 7500 mg (IQR, 5000 to 10125). The median number of treatment cycles for camrelizumab was 12 (IQR, 9 to 16) (Supplemental Table 6).

During CCRT phase, 48 of 49 patients (98.0%) proceeded to the CCRT, and one patient (2.0%) refused radiotherapy (Fig. 1, Supplementary Table 7). A total of 41 patients (41/48, 85.4%) completed 2 cycles of cisplatin and the remaining seven patients (7/48, 14.6%) received one cycle. Two of 48 patients (4.2%) had dose reductions of cisplatin. Overall, the median dose intensity for cisplatin, which was administered every 3 weeks, was 200 mg/m$^2$ (IQR, 200 to 200).

## Efficacy

Overall, all patients (49/49) had an objective response after the completion of induction therapy and before CRT initiated (Table 2 and Supplementary Figs. 1, 2). Among them, 13 patients (26.5%) had a complete response (CR), and 36 (73.5%) had a partial response (PR). After 3 cycles of induction therapy, 28 of 33 patients (84.8%) with available endoscopic biopsy samples from the primary tumors, along with 17 of 34 patients (50%) with available lymph node biopsy samples (39.1% [9/23] after 1 cycle and 72.7% [8/11] after 2–3 cycles) had negative pathological findings (Supplementary Fig. 3). Characteristics were similar between patients who received or refused to be biopsied (Supplementary Table 8). At 12 weeks after CRT, from 48 patients who proceeded to CCRT, 46 (95.8%) patients had CR and one (2.1%) PR, and one died of liver and bone metastases.

By the data cutoff date of May 20, 2023, the median follow-up duration was 28.7 months (range, 22.7–36.7), with 98% (47/48 alive) of the patients followed up for over 24 months. One death due to liver and bone metastases occurred. Both 1-year and 2-year DMFS rates were 98.0% (95% CI, 87.6 to 99.7) (Fig. 2). The 2-year failure-free survival (FFS), OS, and LRFS rates were 95.9% (95% CI, 84.9 to 98.9), 98.0% (95% CI, 87.6 to 99.7), and 97.9% (95% CI, 86.9 to 99.7), respectively (Fig. 2 and Supplementary Fig. 6). The efficacy in per-protocol population is summarized in the Supplementary Table 3.

Of 49 patients, 48 (98.0%) had detectable pre-treatment plasma EBV DNA with a median level of 4710 (IQR, 1670 to 17900) copies/mL (Table 1). The EBV DNA levels became persistently undetectable by the data cutoff date in 95.8% (46/48) of patients, with no relapse events observed. The remaining two patients had persistently detectable plasma EBV DNA, including one with regional PR and another suffering death due to liver and bone metastases (Table 2 and Supplementary Fig. 4).

## Adverse events

All 49 patients were included for evaluating treatment-related adverse events (TRAEs) (Table 3 and Supplementary Table 9). During induction therapy, 23 (46.9%) of the 49 patients had grade 3 or 4 TRAEs. Neutropenia was the most common grade 3 or 4 TRAE (7/49, 14.3%), followed by skin rash (5/49, 10.2%), nausea (5/49, 10.2%), vomiting (3/49, 6.1%), pruritus (3/49, 6.1%), and leukopenia (3/49, 6.1%). Any grade skin

**Table 2 | Response to treatment, pathological evaluation and plasma EBV DNA clearance information**

| | no./total no. (%) |
|---|---|
| **Response to induction phase** | |
| Objective response | 49/49 (100.0) |
| Complete response | 13/49 (26.5) |
| Partial response | 36/49 (73.5) |
| Stable disease | 0 |
| Progressive disease | 0 |
| **Endoscopic biopsy** | |
| **After 3rd cycle of induction phase** | |
| Negative | 28/33 (84.8) |
| Positive | 5/33 (15.2) |
| **Lymph nodes biopsy** | |
| **After 1st cycle of induction phase** | |
| Negative | 9/23 (39.1) |
| Positive | 14/23 (60.9) |
| **After 2nd or 3rd cycle of induction phase** | |
| Negative | 8/11 (72.7) |
| Positive | 3/11 (27.3) |
| **Response to radiation phase**[a] | |
| Objective response | 46/47 (97.9) |
| Complete response | 45/47 (95.7) |
| Partial response | 1/47 (2.1) |
| Stable disease | 0 |
| Disease recurrence | 1/47 (2.1) |
| **Plasma EBV DNA clearance**[b] | |
| Persistently detectable | 2/48 (4%) |
| Change to undetectable | 46/48 (96%) |
| Median (IQR) clearance time, days | 36 (27–56) |
| Range, days | 7–106 |
| Undetectable during induction therapy phase | 41/48 (85%) |
| After 1st cycle of induction therapy | 15/48 (31%) |
| After 2nd cycle of induction therapy | 17/48 (35%) |
| After 3rd cycle of induction therapy | 9/48 (19%) |
| Undetectable during radiotherapy phase | 5/48 (10%) |

*IQR* interquartile range.

[a]Two patients were excluded, including one patient who declined radiotherapy after induction phase and the other patient whose radiotherapy was interrupted prematurely after receiving 9 fractions due to personal reasons. Both of them remained in complete response at the time of the latest follow-up.

[b]There are 48 patients who had positive plasma Epstein-Barr virus (EBV) DNA level at baseline. Quantification testing of plasma EBV DNA level were based on a lower limit of detection cutoff value of 1000 copies/mL. The undetectable levels refer to individuals whose EBV DNA levels are reported at 0 copies/mL.

rash was observed in 27 of 49 (55.1%) patients, which mostly occurred during induction therapy (26/27 [96.3%]). During CCRT, 21 of 48 patients (43.8%) had grade 3 or 4 TRAEs, with the most common being mucositis (14/48, 29.2%).

Through the entire treatment course, 32 of 49 patients (65.3%) had grade 3 or 4 TRAEs, mainly including mucositis (14/49, 28.6%), nausea/vomiting (9/49, 18.4%), neutropenia (8/49, 16.3%), leukopenia (7/49, 14.3%), and anemia (7/49, 14.3%). There were no treatment-related deaths. Regarding late toxicities, 8 of 49 patients (16.3%) had grade 3 or 4 events. Grade 3 or 4 hearing impairment and dry mouth occurred in 5 (10.2%) and 4 (8.2%) patients, respectively.

Forty-eight of 49 patients (98.0%) were reported to have immune-related adverse events (irAEs) (Table 3 and Supplementary Table 10). Most of irAEs were grade 1 or 2. Grade 3 or 4 irAEs occurred in 5 (10.2%) patients, and no immune-related deaths were reported. The most

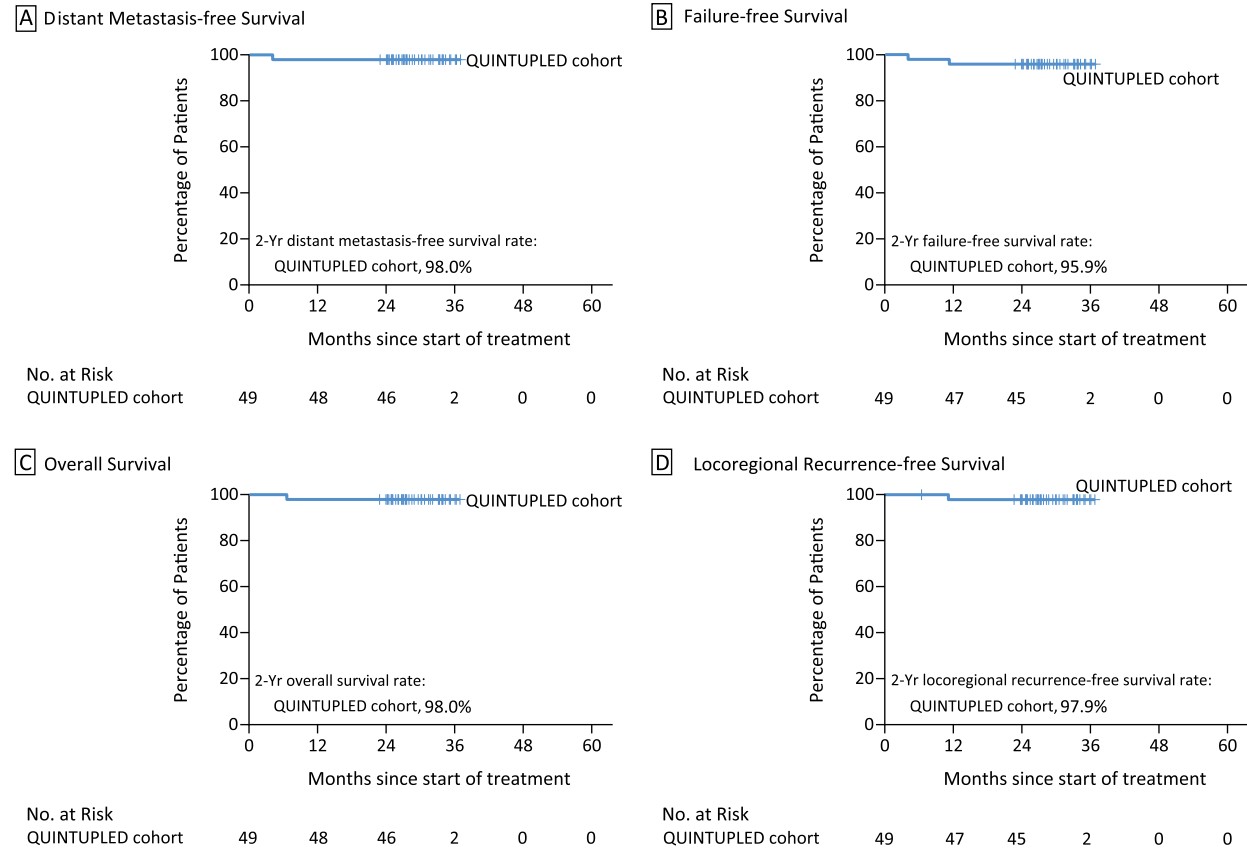

**Fig. 2 | Kaplan–Meier plots of survival outcomes. A** distant metastasis-free survival. **B** Failure-free survival. **C** Overall survival. **D** Locoregional recurrence-free survival (intention-to-treat population). Tick marks indicate censored data.

commonly reported irAEs were reactive cutaneous capillary endothelial proliferation (RCCEP; 42/49, 85.7%), hypothyroidism (15/49, 30.6%) and rash (8/49, 16.3%). RCCEP was observed in 25 (51.0%) patients during the induction therapy, and newly developed in 10 (20.4%) patients during the CCRT and in 7 (14.3%) patients during the maintenance therapy.

## Discussion

This is the first prospective study on the efficacy and safety of a comprehensive treatment, including chemotherapy, immunotherapy, anti-angiogenic therapy and CRT, in patients with local advanced NPC and high risk of distant metastasis. Current findings suggest that the combination of camrelizumab and apatinib with IC of nab-paclitaxel, cisplatin and capecitabine and CCRT has promising distant metastasis control with acceptable safety profile in patients with $T_{any}N3M0$ NPC.

In our study, the 2-year rate of DMFS among N3 patients was significantly higher in comparison to historical cohorts (50%–80%) treated with IC and CCRT[4,9,23–25]. Of note, given that about half of distant metastasis events occurred within 1 year, this new strategy is a promising first-line option for patients with locally advanced NPC and high risk of distant metastasis after validation in large-scale phase 3 trials. In addition, detectable EBV DNA after radical treatment indicated high risk of treatment failure[26,27]. Historically, 50%–75% and 25%–30% of N2-3/N3 patients had detectable EBV DNA levels after IC and CRT, respectively, and subsequently they were the ones with the highest frequency of distant failure[26–28]. In our study, 85% and 96% of the patients had undetectable EBV DNA levels after IC and CCRT, respectively, indicating that this combination regimen did effectively control EBV titer, which is strongly associated with micrometastatic lesions and disease progression.

Patients achieving CR/PR after IC had significantly longer FFS compared to those with stable disease (SD) or progressive disease (PD)[28,29]. The CR rate of 0–8% and SD rate of 10%–16% were observed in N3 patients in previous trials[7,9,28,29]. Our combinatory induction therapy resulted in a CR rate of 26.5% and objective response rate of 100% for patients with N3 disease, indicating that at least 10% of N3 patients who would be classified as SD or PD after treatment with traditional IC, achieve CR or PR from this new combination regimen. In addition, a pathological CR is associated with a survival benefit[30,31]. Indeed, a high pathological CR rate was observed in our study with patients receiving 2–3 cycles of combination regimen, where 85% of biopsied primary tumor samples and 73% of biopsied lymph nodes had negative pathological findings. In the studies involved with traditional CRT, the FFS improvement is mainly derived from LRFS, rather than from DMFS for patients with satisfactory tumor response to IC[28]. This indicates that traditional chemotherapy is not strong enough to eradicate micrometastatic lesions in patients with N3 disease. In fact, the present combination regimen reaches significantly higher FFS and DMFS compared with traditional regimen used in our historical N3 cohort[9], but not LRFS, suggesting that the benefit of FFS in our study mainly originates from distant metastasis control.

In this trial, we used NAB-TPC as IC regimen rather than traditionally used gemcitabine/cisplatin (GC) or cisplatin/5-FU (PF). As we reported recently, TPC as IC regimen is more efficacious than PF for patients with stage IVA to IVB NPC[9], that leads to be a strong recommendation as Class IA evidence by the Chinese Society of Clinical Oncology guidelines. However, none of the N3 patients who received TPC regimen achieved CR with a 2-year FFS rate of only 85%[9]. Therefore, it is reasonable to assume that the addition of the 2 new biological drugs significantly improve the therapeutic efficacy of IC and CCRT.

**Table 3 | Adverse events, according to treatment phase and grade**

| Event | Cumulative TRAEs (n = 49) | | Induction phase (n = 49) | | Concurrent phase (n = 48)[a] | |
|---|---|---|---|---|---|---|
| | Grade 1/2 | Grade 3/4 | Grade 1/2 | Grade 3/4 | Grade 1/2 | Grade 3/4 |
| Any TRAEs | 17 (34.7) | 32 (65.3) | 26 (53.1) | 23 (46.9) | 27 (56.2) | 21 (43.8) |
| Leukopenia | 31 (63.3) | 7 (14.3) | 15 (30.6) | 3 (6.1) | 23 (47.9) | 2 (4.2) |
| Neutropenia | 36 (73.5) | 8 (16.3) | 25 (51.0) | 7 (14.3) | 34 (70.8) | 1 (2.1) |
| Anemia | 41 (83.7) | 7 (14.3) | 43 (87.8) | 0 | 42 (87.5) | 4 (8.3) |
| Thrombocytopenia | 16 (32.7) | 5 (10.2) | 10 (20.4) | 1 (2.0) | 13 (27.1) | 2 (4.2) |
| Hypokalemia | 23 (46.9) | 4 (8.2) | 20 (40.8) | 2 (4.1) | 15 (31.3) | 1 (2.1) |
| Rash | 22 (44.9) | 5 (10.2) | 21 (42.9) | 5 (10.2) | 4 (8.3) | 0 |
| Dry mouth | 40 (83.7) | 7 (14.3) | 0 | 0 | 35 (72.9) | 5 (10.4) |
| Mucositis | 32 (65.3) | 14 (28.6) | 5 (10.2) | 0 | 31 (64.6) | 14 (29.2) |
| Dermatitis | 32 (65.3) | 3 (6.1) | 0 | 0 | 32 (66.7) | 3 (6.3) |
| Nausea | 41 (83.7) | 7 (14.3) | 41 (83.7) | 5 (10.2) | 45 (93.8) | 2 (4.2) |
| Vomit | 32 (65.3) | 6 (12.2) | 26 (53.1) | 3 (6.1) | 34 (70.8) | 3 (6.3) |
| Pruritus | 22 (44.9) | 3 (6.1) | 21 (42.9) | 3 (6.1) | 8 (16.7) | 0 |
| Any late AEs | 39 (79.6) | 8 (16.3) | - | - | - | - |
| Hearing impairment | 20 (40.8) | 5 (10.2) | - | - | - | - |
| Dry mouth | 40 (81.6) | 4 (8.2) | - | - | - | - |
| Trismus | 4 (8.2) | 0 | - | - | - | - |
| Temporal lobe injury | 6 (12.2) | 0 | - | - | - | - |
| Eye damage | 3 (6.1) | 0 | - | - | - | - |
| Peripheral neuropathy | 4 (8.2) | 0 | - | - | - | - |
| Any irAEs | 43 (87.8) | 5 (10.2) | 29 (59.2) | 5 (10.2) | 39 (81.3) | 0 |
| Reactive capillary endothelial proliferation | 42 (85.7) | 0 | 25 (51.0) | 0 | 34 (70.8) | 0 |
| Thyroglobulin decreased | 15 (30.6) | 0 | 11 (22.5) | 0 | 9 (18.4) | 0 |
| Hypothyroidism | 15 (30.6) | 0 | 7 (14.3) | 0 | 3 (6.3) | 0 |
| Anti-thyroid antibody positive | 13 (26.5) | 0 | 8 (16.3) | 0 | 7 (14.3) | 0 |
| Rash | 5 (10.2) | 3 (6.1) | 4 (8.2) | 3 (6.1) | 1 (2.1) | 0 |
| Lipase increased | 5 (10.2) | 0 | 3 (6.1) | 0 | 0 | 0 |
| Diarrhea | 0 | 1 (2.0) | 0 | 1 (2.0) | 0 | 0 |
| Inflammatory cytokine increased | 0 | 1 (2.0) | 0 | 1 (2.0) | 0 | 0 |

[a]Due to one patient declined concurrent chemoradiotherapy after induction phase, there were 48 patients for safety analysis in concurrent phase. Patients may have had more than one event. AEs, Adverse events;

TRAEs, Treatment related adverse events; irAEs, Immune related adverse events.

Meanwhile, we used 2 rather than 3 cycles of cisplatin CRT with the consideration of the reduction of toxicities while maintaining comparable or even achieving improved efficacy with additional agents. Our group has shown that the 2 cycles of once-every-3-weeks concurrent cisplatin is preferred to once-a-week cisplatin in the locoregionally advanced NPC with the advantage of lower acute and late-onset auditory loss toxicities but comparable efficacy[32]. Indeed, the additional two drugs into IC regimen and subsequent CCRT with reduced cisplatin dose reached exceptional efficacy while maintaining low toxicity profile.

With the abovementioned changes, the combination regimen did not result in any unexpected grade 3 or 4 TRAEs or irAEs in the trial. The cumulative TRAE rate of 65% is similar to other studies (ranging from 58% to 76%)[6–9]. Previous studies showed that the proportion of patients completing 3 cycles of IC and 2 cycles of CCRT ranges from 88% to 97% and 88% to 92%, respectively[6,7]. In the present study, 100% (49/49) and 87% (41/47) patients received 3 cycles of IC and 2 cycles of CCRT, respectively. This suggests that the combination regimen is well tolerated. Interestingly, the incidence of RCCEP in our study is higher than that in the CAPTAIN-1st study using combination regimen of camrelizumab plus GC to treat R/M NPC (86% versus 58%)[16], probably because of the multiple-drug interactive effects and intensive monitoring of toxicities developed during the trial in our study with limited sample size.

There are several limitations to our study. First, this trial is a single-arm phase II trial and the conclusions need to be validated in a large randomized, multicenter, controlled trial in the same NPC patient population. Second, this trial was not designed to compare the efficacy of the combination regimen to other treatment options. Although we compared the results to the one from our recent clinical trial cohort[9], further head-to-head comparison in a perspective randomized controlled phase III trial is needed. Third, the question of whether all N3 patients require this combination therapy should be addressed in phase III trials with stratification of disease, e.g., T classification and plasma EBV DNA levels[33,34]. Fourth, specific mechanisms underpinning the efficacy of the QUINTUPLED regimen in targeting metastases remains unclear. Emerging evidence has demonstrated that the antiangiogenic agents not only normalize tumor vascular network, but modulate tumor immune microenvironment through various mechanisms, including promoting T-cell infiltration, inducing M1 macrophage polarization, decreasing the recruitment of Treg and myeloid suppressor cells, and downregulating inhibitory immune checkpoints such as PD-L1, thereby converting the tumor immune microenvironment from immunesuppressive to immune-active[35–38]. These data suggests that the addition of antiangiogenic agents and immunotherapy to standard-of-care IC and CCRT improves antitumoral efficacy.

In conclusion, here, we show that camrelizumab and apatinib combined with IC (nab-paclitaxel, cisplatin and capecitabine) and

CCRT have promising distant metastasis control with acceptable safety profile in stage $T_{any}N3M0$ NPC patients. Larger randomized controlled trials are warranted to validate our findings.

## Methods

### Study design and participants

This was an open-label, single-arm, phase II study conducted at Sun Yat-Sen University Cancer Center (ChiCTR2000032317). The first and last patients were officially enrolled in cohort 1 on 26 May 2020, and 21 July 2021, respectively. The trial protocol was approved by the Chinese Ethics Committee of Registering Clinical Trials. This study was conducted in accordance with the Declaration of Helsinki and the International Standards of Good Clinical Practice. Eligible patients aged 18–65 years with previously untreated, histologically proven non-keratinising stage $T_{any}N3M0$ NPC (American Joint Committee on Cancer, 8th edition) were enrolled. All patients had an Eastern Cooperative Oncology Group performance status of 0 or 1, and adequate bone marrow, liver, and renal function. Key exclusion criteria included a history of previous radiotherapy, chemotherapy, or surgery to the primary tumor or lymph nodes; any severe coexisting diseases; treatment with palliative intention; previous malignancy; pregnancy or lactation; a history of protocol-specified autoimmune disease. The full eligibility criteria are available in the study protocol (available in the Supplementary Information file). All patients provided informed consent.

### Procedures

All eligible patients received three 21-day cycles of IC (NAB-TPC, nab-paclitaxel 200 mg/m² intravenously on day 1, cisplatin 60 mg/m² intravenously on day 1, and capecitabine 1000 mg/m² orally twice daily on days 1 to 14), followed by CCRT (cisplatin 100 mg/m² intravenously on day 1 of each cycle for 2 cycles concurrently with IMRT). The radiotherapy protocol is provided in the Supplementary Information file. For the planning target volumes of GTVnx, GTVnd, $CTV_{high-risk}$, and $CTV_{low-risk}$, total radiation doses of 68–72 Gy, 64–68 Gy, 60-64 Gy, and 54-58 Gy, respectively, were administered in 30–33 fractions, delivered daily at five fractions per week.

Camrelizumab was administrated intravenously at a fixed dose of 200 mg every 3 weeks starting from the first cycle of IC until 12 months. During the induction phase, all patients simultaneously received oral apatinib from the first day of IC (Dose of 250 mg per day from day 1 to day 56 in the first version of the protocol). The schedule was modified to be 5 days on, followed by 2 days off every week; (A total of 3750 mg per cycle in second version of protocol given the safety concerns) for 8 weeks (It was recommended that patients stopped apatinib at least 7 days before starting IMRT).

Tumor response was evaluated according to the Response Evaluation Criteria in Solid Tumors (RECIST; version 1.1) using nasopharyngeal and neck MRI at 1 week after IC and 12 weeks after CRT were completed. The follow-up restaging images with additional whole-body CT were performed every 3 months for 3 years, every 6 months for 5 years, and annually thereafter. Patients underwent routine check-ups weekly during the IC and CCRT, and every 3 weeks during the maintenance therapy. Patients were taken off from the study if they had tumor progression or severe comorbidities developed during treatment or withdrew consent at any time during the study.

Acute toxicities throughout the treatment period were recorded and graded in accordance with the National Cancer Institute Common Terminology Criteria for Adverse Events (version 5.0). Late radiotherapy toxicities were assessed by means of the Late Radiation Morbidity Scoring Criteria of the Radiation Therapy Oncology Group.

### Outcomes

The primary endpoint was DMFS, defined as the time from the initiation of induction therapy to the first documented distant metastasis. Secondary endpoints were overall survival (OS), FFS, locoregional recurrence-free survival (LRFS), tumor response, and toxicity profile (Supplementary information). Patients who were lost to follow-up or were still alive without distant metastasis or locoregional recurrence at the end of the trial had their data censored at the date of last follow-up.

### Statistical analysis

Fleming's one-sample, multiple-testing procedure was adopted for the sample size calculation with an α of 0.05 and a β of 0.18. Per-protocol analysis was reported in the Supplementary Information file (Supplementary Tables 2, 3). Characteristics of historical cohort with N3 disease are summarized in Supplementary Table 4. The 12-month DMFS rate of ≤76% was considered unacceptable, which was based on the previous results in our center[4]. The 12-month DMFS rate with the study treatment was expected to reach 90%, and 46 evaluable patients were required. Considering a dropout rate of 8%, the total sample size was 50 patients.

The intention-to-treat analysis set included all eligible patients who received at least one dose of the study treatment. We estimated time-to-event data using the Kaplan–Meier method and calculated 95% CIs using Greenwood's formula. Data were summarized as frequencies and proportions for categorical variables and as medians and ranges for continuous variables. Analyses were conducted with the use of Stata software, version 14.2 (StataCorp).

### Reporting summary

Further information on research design is available in the Nature Portfolio Reporting Summary linked to this article.

## Data availability

Source data are not publicly available due to involving patient privacy, but can be accessed on request from the corresponding author for 10 years. Any request should be sent to the corresponding author, Y.-Q.X., via email at xiangyq@sysucc.org.cn, along with a detailed description of your research protocol; individual deidentified participant data will be shared. The corresponding author and Sun Yat-sen University Cancer Center will evaluate the reasonability of the request for our data and reserve the right to decide whether to share the data or not. And the data is only used for the research purpose. This process requires an average of 6 months. All data shared will be de-identified and will be available for 1 year after access is granted. The study protocol is available as Supplementary Note in the Supplementary Information file. The remaining data are available within the Article, Supplementary Information.

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

## Acknowledgements

Supported by grants from the National Natural Science Foundation of China (82002855, 82202905 and 82203755), Guangzhou Basic and Applied Basic Research Project (No. 2023A04J2142, 2023A04J2136) and

China Postdoctoral Science Foundation (2022M713593). We thank the patients who participated in this study, their families, and the medical, nursing, and research staff at the study center. The sponsor(s) had no role in study design, data collection and analysis, or manuscript writing.

## Author contributions

Drs H Liang and Y-Q Xiang had full access to research data in the study and took responsibility for the integrity of the data and the accuracy of the data analysis. Drs H Liang, Y-F Jiang, G-Y Liu, L. Wang and J-W Wang contributed equally to this study. Drs Y-Q Xiang, C Xie and C Zhao were joint senior authors. Concept and design: Y-Q Xiang, C Xie and C Zhao. Acquisition, analysis, or interpretation of data: H Liang, Y-F Jiang, G-Y Liu, L. Wang, J-W Wang, N Lu, W-X Xia, L-R Ke, Y-F Ye, J-L Duan, W-X Bei, S-H Dong, W-Z Li and L-T Liu. Drafting of the manuscript: H Liang, Y-F Jiang, G-Y Liu, L. Wang and J-W Wang. Critical revision of the manuscript for important intellectual content: Y-Q Xiang, C Xie and C Zhao. Statistical analysis: H Liang, Y-F Jiang, G-Y Liu, L. Wang, J-W Wang, N Lu, W-X Xia, L-R Ke, Y-F Ye, J-L Duan, W-X Bei, S-H Dong, W-Z Li and L-T Liu. Obtained funding: Y-Q Xiang, G-Y Liu and H Liang. Administrative, technical, or material support: Y-Q Xiang, C Xie and C Zhao. Supervision: Y-Q Xiang, C Xie and C Zhao.

## Competing interests

The authors declare no competing interests.

## Additional information

[1]Department of Nasopharyngeal Carcinoma, Sun Yat-Sen University Cancer Center, Guangzhou, China. [2]State Key Laboratory of Oncology in South China, Collaborative Innovation Center of Cancer Medicine, Guangdong Key Laboratory of Nasopharyngeal Carcinoma Diagnosis and Therapy, Guangzhou, China. [3]Department of Oncology, Sun Yat-Sen Memorial Hospital, Sun Yat-Sen University, Guangzhou, China. [4]Department of Ultrasound, Sun Yat-Sen University Cancer Center, Guangzhou, China. [5]Department of Medical Imaging, Sun Yat-Sen University Cancer Center, Guangzhou, China. [6]Clinical Research Design Division, Sun Yat-Sen Memorial Hospital, Sun Yat-Sen University, Guangzhou, China. [7]Department of Pathology, Sun Yat-Sen University Cancer Center, Guangzhou, China. [8]Thoracic and GI Malignancies Branch, National Cancer Institute, National Institutes of Health, Bethesda, USA. [9]These authors contributed equally: Hu Liang, Yao-Fei Jiang, Guo-Ying Liu, Lin Wang, Jian-Wei Wang. ✉e-mail: zhaochong@sysucc.org.cn; changqing.xie@nih.gov; xiangyq@sysucc.org.cn

