## [Peer Review File · Nature Communications]

Camrelizumab and apatinib plus induction chemotherapy and concurrent chemoradiotherapy in stage N3 nasopharyngeal carcinoma: a phase 2 clinical trialEditorial Note: Parts of this Peer Review File have been redacted as indicated to remove third-party material where no permission to publish could be obtained.

REVIEWER COMMENTS

Reviewer #1 (Remarks to the Author): with expertise in nasopharyngeal cancer, (immuno)therapy

This is an interesting single-arm phase 2 trial of a quintet combination of induction taxane, cisplatin, capecitabine, apatinib, and camrelizumab, followed by cisplatin-radiotherapy, and maintenance camrelizumab for 1 year in exclusively patients with N3 nasopharyngeal carcinoma (NPC). Rationale by the authors was that this subset of NPC patients fared poorly even with standard-of-care (SOC) induction chemotherapy and chemo-radiotherapy, although admittedly, SOC for this group of patients in 2023 would be induction chemotherapy, chemo-radiotherapy, and maintenance metronomic capecitabine for 1 year. Primary endpoint of this present study was 1-y distant metastasis-free survival (DMFS). The investigators recruited 49 patients, all of whom were eligible for safety analysis, and 48 were eligible for efficacy analysis (1 patient declined chemo-radiotherapy). They found that 2-year survival outcomes were >90% across all endpoints, with 1 death attributable to NPC; this is also a patient with rapid relapse to the bones and liver. They thus concluded that this is a “feasible” and “effective” regimen for this group of patients who are at high risk of distant metastases.

While the study is interesting, I have the following major comments for consideration:

1. Abstract (Line 50-52): The statement of “Immunotherapy combination with standard treatment regimen was reported benefit in nasopharyngeal carcinoma (NPC), however, N3 patients did not show the benefit” is contentious – where did the author draw this evidence from?

To date, there are only 2 trials of immunotherapy (checkpoint blockade) reported in NPC; CONTINUUM is a phase 3 trial investigating sintilimab in patients with N2-3 and T4N+ NPC, and NEOSPACE is a phase 2 trial investigating pembrolizumab exclusively in T4 and N3 NPC. CONTINUUM was reported as a positive trial at the ASCO Annual meeting in June this year,

although the results have not yet been published. Likewise, NEOSPACE reported ~70% 2-y DFS rate in T4 and N3 NPC, which may not represent an improvement to current SOC. Thus, the statement by the authors is flawed.

2. Background: There are a few statements that require revisions.

a. (Lines 72-73): It is unclear on the validity of the comment about MRI improving local control, although certainly the incorporation of MRI helps to improve delineation.

b. (Lines 78-81): The comment on large phase III trials with induction chemotherapy and chemo-radiotherapy not improving DMFS is untrue and contentious! The separation of the curves was certainly apparent in the seminal trials by Sun Ying, et al. *Lancet Oncology*, 2016 and Zhang Yuan, et al. *NEJM*, 2019, except that these trials were not powered for DMFS as a primary endpoint, and thus, the difference may not be statistically significant.

3. Statistical considerations: This reviewer noted the use of 1-y DMFS of 76% as a benchmark for N3 NPC for the calculation of power and sample size of this study. However, the publication referenced was dated in 2014, and this precedes all the contemporary trials of induction chemotherapy and chemo-radiotherapy. Thus, one would query the legitimacy of using this benchmark – a 2-y DMFS would have been more reasonable.

a. A 1-y DMFS endpoint would be influenced by several factors, including (1) the interval of imaging (note that staging investigations were only done annually); (2) whether there was onset of locoregional recurrences during the 1 year, which would have led to censoring of patients ahead of their 1 y assessment for DMFS; and (3) most importantly, the length of treatment relative to the timing of the endpoint – see Comment #5.

b. On this note, this reviewer noted that staging at 1 year was done using chest radiograph and abdominal scan. Admittedly, these modalities would not be sensitive for the detection of distant metastases compared with full body CT or FDG-PET-CT. Could this have led to underreporting of events?

4. Treatment procedure and compliance: When reviewing the treatment procedures, there were differences between the present trial and the prior induction chemotherapy trial of triplet versus doublet chemotherapy by the same group (Wang Zhong Li, et al. *JAMA Oncol*, 2022). Specifically, here, they used 3 cycles of TPX, while in the preceding trial, the

investigators used 2 cycles of TPX, and here, they replaced taxol with abraxane. Additionally, they incorporated apatinib to TPX during the induction phase, and camrelizumab from induction throughout adjuvant phases of treatment; this would also imply that they have now investigated a 5-drug regimen for induction chemotherapy for high-risk NPC.

a. How were these patients staged? Was it N3 by 7th or 8th edition? This was not mentioned in the main text.

b. With this regimen, the authors reported a 26.5% complete response rate post-induction, but I am unsure about the tolerability of this 5-drug combo (see Comment #7).

c. What was the time to starting adjuvant camrelizumab? Was there any dose reduction or treatment interruption? Did adjuvant camrelizumab delay recovery from radiotherapy toxicities?

d. Perhaps more importantly, what was the scientific rationale for this quintet combination? Apatinib, which is a VEGFR2 small molecule inhibitor, has single agent activity in recurrent-metastatic NPC (Luo, et al. Oral Oncology, 2021). However, was there phase 1b data supporting the safety of this combination? This information was lacking in the Introduction and reporting of the results.

5. Results (DMFS):

a. Choice of DMFS as the primary endpoint: This reviewer queries the choice of this endpoint at 1-y mark, especially when the entire treatment duration was longer than 1 year.

b. How did they estimate the time to event? Was it from the start of treatment? If so, then this reviewer would contest that the choice of a 1-y endpoint is flawed, since it is highly unlikely for patients to progress within such a short window. It would be difficult to judge if this regimen was indeed “effective” in eradicating the occult distant metastases. At least, the authors reported on the 2-year survival rates.

6. Results (Response to induction treatment): I have the following comments on the evaluation of response to induction treatment.

a. Foremost, there seems to be a discordance between radiological response and pathological response (73.5% PR by RECIST but 84.8% pathological -ve biopsy). Could this be due to a sampling error?

b. This reviewer is also intrigued by the modest CR rate of 26.5%, even with a quintet

combination. With such an intensive regimen, one might have expected a higher CR rate. To interpret the data better, more information ought to be included in Table 1. For example, were the nodes bulky? How many levels were involved? What was the median gross tumour volume for the primary and nodal lesions? See my Comment #4a on stage classification.

c. Perhaps, it would have been very useful to have EBV DNA response data to the induction phase. What was the proportion of patients who had a complete clearance by the end of induction treatment? What was the magnitude of drop in EBV DNA counts?

d. Overall, this reviewer is unconvinced that the pathological responses are indicative of efficacy to this quintet combo. Moreover, conventional imaging, unless PET-CT, is not ideal for assessing CR of nodes; as a case in point, a lymph node of 3 cm size shrinking to 0.8 cm would be considered as PR? However, it could well be a pathological CR, since involved lymph nodes don't typically shrink entirely even months post-treatment (Li WF, et al. Oral Oncology, 2017).

If anything, EBV DNA and PET post-induction would have been highly informative of efficacy, especially since the concern with patients with N3 NPC is the risk of distant metastases – thus, the current measures do not offer insights in this regard.

e. Perhaps, the authors could include some pre- and post-induction images of extreme responders to the induction treatment, which would be informative to the readers.

7. Results – Figure 2: While the data compared with the historical cohort looks promising, such an analysis is flawed. Was the historical cohort matched in some way? Otherwise, this reviewer will urge the investigators to remove the historical cohort as its inclusion will lead to misinterpretation.

8. Toxicity results: The incidence of 65.3% (32 of 49) patients for G3-4 treatment-related adverse events is substantial. This reviewer would not consider such a regimen to be “well-tolerated”. Moreover, the incidence of 46.9% (23 of 39) patients with G3-4 adverse events during induction chemotherapy is substantial, and this reviewer is surprised that dose intensities for the chemotherapeutic agents remained at 100%?

9. Discussion: There need to be some elaboration on the mechanisms underpinning the efficacy of this regimen in targeting occult metastases in NPC.

Reviewer #2 (Remarks to the Author): with expertise in nasopharyngeal cancer, (immuno)therapy

1. Needs editing to improve English fluency.
2. Why have the investigators selected a relatively obscure IC regimen to build on? Most use either GC or TPF lite. This makes interpretation of these results a bit challenging.
3. It is unclear that data were behind the modification during the study of apatinib dosing from continuous to just 5 d/ week.
4. It is surprising that tumor response assessment did not include PET CT nor ct EBV DNA. PET CT in particular is SOC many places for the evaluation 12-16 weeks post XRT and having only CT and MRI could lead to significant underdetection of persistent or recurrent disease.
5. Were PD-L1 TPS or CPS data collected on tumors?
6. It seems that the timing of biopsies on IC was not consistent. Some after one, some after two cycles, and only a fraction of the 49 patients had either biopsies.
7. In the results, ctEBVDNA are reported. See item 4 above. the prespecified plan and ct EBV DNA collection and evaluation methods should be specified. Or was this as hoc? How is “ persistently” negative ct EBV DNA defined?
8. Do the authors have a specific reason for having chosen camrelizumab for which reactive cutaneous capillary endothelial proliferation seems to be a challenge?
9. Since the IC backbone is not standard , and there are two new agents added at the same time, what should plans be for discerning contribution of each new component of this treatment to the promising outcome?
10. I would discourage any graphs showing survival data of non – randomized non-concurrently rerated patients . This over represents the meaningfulness of the comparison.
11. It is unclear from the attached protocol how tumor volumes and LN volumes for radiation were determined. While not a primary endpoint issue, with all of these chemoradiation studies going on in NPC, it is essential for the community to understand how RT volumes , when tumor and LN volumes are changing with IC, are being addressed. Are all plans based upon initial pre IC staging and not modified at all based upon responses to IC? Or are plans based on some modification based upon IC response?

Reviewer #3 (Remarks to the Author): with expertise in nasopharyngeal cancer, (immuno)therapy

This manuscript describes an open-label, single-arm, phase II study evaluating the efficacy and toxicities of camrelizumab, apatinib plus induction chemotherapy and concurrent chemoradiotherapy in stage N3 nasopharyngeal carcinoma. Overall, the quality of the manuscript is good, it clearly describes the objectives, design and results of the study.

Comments:

In the study, the investigators used NAB-TPC as induction chemotherapy regimen rather than gemcitabine and cisplatin, which is the standard of care for locoregionally advanced nasopharyngeal carcinoma. The authors should explain why. Is there any evidence supports that NAB-TPC is more efficacious than gemcitabine and cisplatin as induction chemotherapy?

In a previous phase II study evaluating camrelizumab plus apatinib in patients with recurrent or metastatic nasopharyngeal carcinoma, investigators reported that the most common complication was nasopharyngeal necrosis (9/16; 56.3%) (J Clin Oncol. 2023 May 10;41(14):2571-2582). How many patients in the present study have nasopharyngeal necrosis? The schedule of apatinib has been modified in second version of protocol given safety concern in the present study. The safety concern should be clarified.

Reviewer #4 (Remarks to the Author): with expertise in clinical trial study design, biostatistics

Thank you for submitting your manuscript. It is a well-organized manuscript and I have a few comments that you may want to consider:

(1). In the background, it's clear that there's a significant issue with distant metastasis in N3 disease patients. However, for greater emphasis and clarity, you might consider reiterating

the main problem statement. This will give readers a clearer sense of the problem's importance.

(2). The predetermined values (like DMFS rates of $\leq 76\%$ being considered unacceptable) are specific but might need justification. Why was this particular percentage chosen as the cutoff? This clarity can help readers understand the context better.

(3). At line 164, a minor typo exists: it should likely be "StataCorp" instead of "StateCorp."

(4). At line 168, "Of them, 1 patient whose diagnosis did not meet eligibility criteria were excluded from the study". It would be more grammatically correct to state "was excluded" instead of "were excluded".

(5). The interpretation of certain statistics could be made more explicit. For example, while p-values are not mentioned in the results, as they were included in Figure 2, ensuring their interpretation is clear would be beneficial.

(6). In the discussion section, the author mentioned that the DMFS rate improvement of 11.7% is presented without context. There should be a discussion on the clinical significance of this improvement.

(7). As an open-label, single-arm, phase II study, the results might be prone to biases. Was the study population diverse enough to ensure the results were generalizable?

Responses to Reviewer #1

Comment 1. Abstract (Line 50–52): The statement of “Immunotherapy combination with standard treatment regimen was reported benefit in nasopharyngeal carcinoma (NPC), however, N3 patients did not show the benefit” is contentious - where did the author draw this evidence from?

To date, there are only 2 trials of immunotherapy (checkpoint blockade) reported in NPC; CONTINUUM is a phase 3 trial investigating sintilimab in patients with N2–3 and T4N+ NPC, and NEOSPACE is a phase 2 trial investigating pembrolizumab exclusively in T4 and N3 NPC. CONTINUUM was reported as a positive trial at the ASCO Annual meeting in June this year, although the results have not yet been published. Likewise, NEOSPACE reported ~70% 2-y DFS rate in T4 and N3 NPC, which may not represent an improvement to current SOC. Thus, the statement by the authors is flawed.

Response: We thank the reviewer for your expertise comments. We agree with the comments about CONTINUUM and NEOSPACE trials. However, as we discuss in the response to comment #2b, reported CONTINUUM trial has shown that immune checkpoint inhibitor (sintilimab) in combination with standard-of-care regimen with sintilimab maintenance failed to significantly improve progression-free survival in N2 and N3 disease. While NEOSPACE trial included over 40% N1-2 patients though they did show 2y DFS of 70% and 2y DMFS 80%. Therefore, we have revised this sentence accordingly that “patients with N3 nasopharyngeal carcinoma have a high risk of distant metastasis, however, if the addition of immunotherapy to standard treatment could improve efficacy is unclear.”

Comment 2. Background: There are a few statements that require revisions.

a. (Lines 72–73): It is unclear on the validity of the comment about MRI improving local control, although certainly the incorporation of MRI helps to improve delineation.

b. (Lines 78–81): The comment on large phase III trials with induction chemotherapy and chemo-radiotherapy not improving DMFS is untrue and contentious! The separation of the curves was certainly apparent in the seminal trials by Sun Ying, et al. *Lancet Oncology*, 2016 and Zhang Yuan, et al. *NEJM*, 2019, except that these trials were not powered for DMFS as a primary endpoint, and thus, the difference may not be statistically significant.

Response: We appreciate reviewer’s expertise input.

For point 2a:

We have amended statement in revised manuscript that “The application of intensity-modulated radiotherapy (IMRT) and combined chemoradiotherapy has substantially improved locoregional control.”.

For point 2b:

We acknowledge the nonsignificant difference may be derived from insufficient power because of limited sample size in their study. But more noteworthy, a subgroup analysis in the Supplementary appendix included in the paper (*Zhang et al, N Engl J Med 2019*) suggested the existence of interaction effects between different N category, that we personally discussed with our colleague Dr. Zhang Yuan (the first author of *Zhang et al, N Engl J Med 2019*) and confirmed this interaction.

This interaction has strongly indicated that the N3 patients belong to a special group among NPC patients in their trial and there was no evidence indicating the N3 patients would benefit from GP regimen. Furthermore, in 2023 ASCO annual meeting, Dr. Jun Ma clearly reported significant heterogeneity in N stage of CONTINUUM study (**As shown in the following Figure 1 of slide screenshot; Ma J, et al. 2023 ASCO abstract LBA6002, Slide #18**). Specifically, the sintilimab group (sintilimab + GP induction therapy followed by CCRT and sintilimab maintenance up to total of 12 cycles) failed to significantly improve progression-free survival in advanced N stage, including N2 and N3 disease. Therefore, up to now, it is still uncertain whether the available treatment options are effective for N3 patients so far. Our group is the first to report that the TPC regimen is effective on NPC patients (*Li WZ, et al. JAMA Oncology 2022; Liu GY, et al. JAMA Oncology 2022*), and the TPC regimen has been strongly (Level I) recommended as Class IA evidence by CSCO guideline. To improve the prognosis of N3 patients, we optimized the TPC regimen as well as added antiangiogenic therapy and immunotherapy, designed as current QUINTUPLED regimen. And our results show that the QUINTUPLED regimen has promising distant metastasis control with acceptable safety profile in N3 NPC patients. In addition, thanks for reviewer's comment again, we have revised the statement as the following at Page 5 in the revised manuscript: Although treated with IC and CCRT, distant metastasis-free survival (DMFS) in patients with N3 disease still remains unsatisfactory, as reported by several phase III clinical trials, even with modification of IC regimen or adjuvant metronomic capecitabine.

Figure 1 (Slide screenshot). Heterogeneity in N category.

[FIGURE REDACTED]

Reference:

1. Zhang Y, Chen L, Hu GQ, et al. Gemcitabine and Cisplatin Induction Chemotherapy in Nasopharyngeal Carcinoma. *The New England journal of medicine* 2019; 381:1124-35.
2. Ma J, Sun Y, Liu X, et al. PD-1 blockade with sintilimab plus induction chemotherapy and concurrent chemoradiotherapy (IC-CCRT) versus IC-CCRT in locoregionally-advanced nasopharyngeal carcinoma (LANPC): A multicenter, phase 3, randomized controlled trial (CONTINUUM). *Journal of Clinical Oncology* 2023;41: LBA6002-LBA.

3. Li WZ, Lv X, Hu D, et al. Effect of Induction Chemotherapy with Paclitaxel, Cisplatin, and Capecitabine vs Cisplatin and Fluorouracil on Failure-Free Survival for Patients with Stage IVA to IVB Nasopharyngeal Carcinoma: A Multicenter Phase 3 Randomized Clinical Trial. *JAMA oncology* 2022; 8:706-14.
4. Liu GY, Li WZ, Wang DS, et al. Effect of Capecitabine Maintenance Therapy Plus Best Supportive Care vs Best Supportive Care Alone on Progression-Free Survival Among Patients with Newly Diagnosed Metastatic Nasopharyngeal Carcinoma Who Had Received Induction Chemotherapy: A Phase 3 Randomized Clinical Trial. *JAMA oncology* 2022; 8:553-61.

Comment 3. Statistical considerations: This reviewer noted the use of 1-y DMFS of 76% as a benchmark for N3 NPC for the calculation of power and sample size of this study. However, the publication referenced was dated in 2014, and this precedes all the contemporary trials of induction chemotherapy and chemoradiotherapy. Thus, one would query the legitimacy of using this benchmark - a 2-y DMFS would have been more reasonable.

a. A 1-y DMFS endpoint would be influenced by several factors, including (1) the interval of imaging (note that staging investigations were only done annually); (2) whether there was onset of locoregional recurrences during the 1 year, which would have led to censoring of patients ahead of their 1 y assessment for DMFS; and (3) most importantly, the length of treatment relative to the timing of the endpoint - see Comment #5.

b. On this note, this reviewer noted that staging at 1 year was done using chest radiograph and abdominal scan. Admittedly, these modalities would not be sensitive for the detection of distant metastases compared with full body CT or FDG-PET-CT. Could this have led to underreporting of events?

Response: Thanks for your expert input. At the time of design of this study in 2019, several studies focusing on N3 disease were identified and summarized in the following **Table 1**. There were several common points observed from these studies. Firstly, the majority of N3 patients received high-intensity treatment, including induction/adjuvant therapy and CCRT. Secondly, distant metastasis is the most important determinant of the survival rates for N3 patients despite the availability of aggressive treatment strategies of the combination of chemotherapy and radiation therapy. Thirdly, about half of distant metastasis events occurred within 1 year. Fourthly, distant metastasis control rates were similar but unsatisfactory. Although the study reference originated from our center was dated in 2014 (*Sun X, et al. Radiology and Oncology 2014*), the patients with N3 disease who are treated in our center usually receive high-intensity treatment, i.e., neoadjuvant chemotherapy in combination with concurrent chemotherapy (current standard therapy). Given the quality of the literature (decided by NOS, as shown in the NOS column of Table 1) and much more reliable and detailed data source from the cohorts in our cancer center, we eventually decide the sample size based this reference. Meanwhile, the previous study showed that there was no obvious change in the DMFS rate over the past two decades even with the advances in radiation procedure, image technology, and chemotherapy regimens (*Sun XS, et al. Int J Radiat Oncol Biol Phys 2019*). In addition, with the follow-up data (median follow-up duration is 28.7 months, range from 22.7 to 36.7), only 1 event of distant metastasis and recurrence was observed. Based on the excellent distant

control rates with long-term follow up time and historical rates, the assumed sample size would be smaller under the assumption of constant type I and II errors, which strongly indicates that our sample size is adequate. Meanwhile, both 2- and 3-year DMFS were pre-set as part of secondary endpoints. With the median follow-up duration of 28.7 months (range, 22.7 to 36.7), the 98% (47/48 alive) patients were followed up for over 24 months. Furthermore, here, we reported 2-year DMFS which is sufficiently representative of long-term efficacy as indicated based on previous reports (Tang LQ, et al. *Lancet Oncology* 2018; Rotolo F, et al. *J Natl Cancer Inst* 2017). Our results show that the QUINTUPLED regimen has promising disease control in N3 NPC patients. (Supplementary Figure S6 in revised Appendix and Figure 1-2 shown in this response; Li WZ, et al. *JAMA Oncology* 2022; Ma J, et al. 2023 ASCO abstract LBA6002; Liu LT, et al. *Lancet Oncology* 2023).

Table 1. Summary of trials for N3 NPC patients before design of the present trial.

Year	Area	RT	IC	Cases	No. of N3	DMFS rates	NOS*	Note
2010	Hong Kong ¹	IMRT	Yes	173	15	67.3% (2 y)	8	
2011	Guang Zhou ²	2DRT	Yes	764	232 (N2-3)	65.4% (5 y)	8	
		IMRT	Yes	512	169 (N2-3)	60.9% (5 y)		
2013	Guang Zhou ³	IMRT	Yes	198	26	50.1% (5 y)	7	
2014	Guang Zhou⁴	IMRT	Yes	868	29	62.1% (2 y)	8	76% (1 y)
2017	Fu Zhou ⁵	IMRT	Yes	22	22	81.8% (3 y)	6	WHO II 32%
2017	Nan Jing ⁶	IMRT	Yes	614	40	62.5% (2 y)	7	80% (1 y)
2018	Hang Zhou ⁷	IMRT	Yes	110	110	65% (5 y)	3	Low quality
2019	Fu Zhou ⁸	IMRT	Yes	130	109	70.3% (3 y)	6	~88% (1 y) WHO II 9.2%

*Newcastle-Ottawa Scale score is used for assess the quality, including high quality for 7-9 score, middle quality for 4-6, and lower than 4 for low quality.

Distant metastasis predominates as the main pattern and the most lethal mode of treatment failure in patients with N3 disease (Sun XS, et al. *Int J Radiat Oncol Biol Phys* 2019; Li WZ, et al. *JAMA Oncology* 2022; Liu LT, et al. *Lancet Oncology* 2023). In order to directly test whether the new regimen reduces the distant failure rate, we conducted this phase II study with the primary endpoint of DMFS. Meanwhile, the DFS was one of secondary endpoints. In fact, our results show that the QUINTUPLED regimen has promising disease control in N3 NPC patients (Supplementary Figure S6 in revised Appendix and Figure 1-2 shown in this response; Li WZ, et al. *JAMA Oncology* 2022; Ma J, et al. 2023 ASCO abstract LBA6002; Liu LT, et al. *Lancet Oncology* 2023), but higher LRFS was observed, indicating that the benefit of PFS in our study mainly originates from distant metastasis control.

For the onset of locoregional recurrence during 1 year, the event would be marked as locoregional failure event, is not the censoring for the endpoint of DMFS. In other words, the patient who occurred locoregional recurrence within 1 year would not be censored as a distant metastasis event, but if distant metastasis is subsequently developed, the patient would be considered as a positive event by the time of distant metastasis detected.

We are sorry for the confusion of our wording in terms of re-staging. As described in our manuscript, patients underwent routine blood tests and blood biochemistry weekly during the phase of induction chemotherapy and chemoradiotherapy, and every 3 weeks during the maintenance therapy phase. Nasopharynx and neck MRI and nasopharyngeal endoscopy will be performed before and after induce chemotherapy and at the end of radiotherapy. After radiation therapy finished, routine follow-up evaluations (physical examination, blood examination, nasopharyngeal endoscopy, nasopharynx and neck MRI, chest CT, and abdominal CT) were routinely performed every 3 months until 3 years, every 6 months until 5 years, and annually thereafter. The restaging evaluation was done at every routine follow-up timepoint as abovementioned, not once a year! To avoid misunderstandings, we have revised the description. And for the economic factors, the chest and abdominal CT is optimal but not mandatory. Indeed, the follow up strategy of our study fully compliant with CSCO guideline, and almost eligible patients in this trial received chest and abdominal CT as their chest and abdominal examination. To date, all alive patients are being followed up and we are in constant communication with them and their families.

Figure 1 (for Reviewer #1 Comment #3): The comparison of DMFS results among reported clinical trials. Among these, the curve from sintilimab + GP + CCRT trial is derived from all sintilimab + GP + CCRT group patients combined which included only 34% N3, but 66% N1-2 patients.

Figure 2 (for Reviewer #1 Comment #3): The comparison of PFS results among reported clinical trials. Among these, the curve from sintilimab + GP + CCRT trial is derived from all sintilimab + GP + CCRT group patients combined which included only 34% N3, but 66% N1-2 patients.

Reference:

1. Wong FC, Ng AW, Lee VH, et al. Whole-field simultaneous integrated-boost intensity-modulated radiotherapy for patients with nasopharyngeal carcinoma. *International journal of radiation oncology, biology, physics* 2010;76:138-45.
2. Lai SZ, Li WF, Chen L, et al. How does intensity-modulated radiotherapy versus conventional two-dimensional radiotherapy influence the treatment results in nasopharyngeal carcinoma patients? *International journal of radiation oncology, biology, physics* 2011;80:661-8.
3. Sun X, Zeng L, Chen C, et al. Comparing treatment outcomes of different chemotherapy sequences during intensity modulated radiotherapy for advanced N-stage nasopharyngeal carcinoma patients. *Radiation oncology (London, England)* 2013;8:265.
4. Sun X, Su S, Chen C, et al. Long-term outcomes of intensity-modulated radiotherapy for 868 patients with nasopharyngeal carcinoma: an analysis of survival and treatment toxicities. *Radiation oncology: journal of the European Society for Therapeutic Radiology and Oncology* 2014; 110:398-403.
5. Zhang Y, Chen M, Chen C, Kong L, Lu JJ, Xu B. The efficacy and toxicities of intensive induction chemotherapy followed by concurrent chemoradiotherapy in nasopharyngeal carcinoma patients with N3 disease. *Scientific reports* 2017; 7:3668.
6. Wu LR, Liu YT, Jiang N, et al. Ten-year survival outcomes for patients with nasopharyngeal carcinoma receiving intensity-modulated radiotherapy: An analysis of 614 patients from a single center. *Oral oncology* 2017; 69:26-32.
7. Chen J, Liu T, Sun Q, Hu F. Clinical and prognostic analyses of 110 patients with N3 nasopharyngeal carcinoma. *Medicine* 2018;97: e13483.
8. Zong J, Xu H, Chen B, et al. Maintenance chemotherapy using S-1 following definitive chemoradiotherapy in patients with N3 nasopharyngeal carcinoma. *Radiation oncology (London, England)* 2019;14:182.
9. Sun X, Su S, Chen C, et al. Long-term outcomes of intensity-modulated radiotherapy for 868 patients with nasopharyngeal carcinoma: an analysis of survival and treatment toxicities. *Radiation oncology: journal of the European Society for Therapeutic Radiology and Oncology* 2014; 110:398-403.
10. Sun XS, Liu SL, Luo MJ, et al. The Association Between the Development of Radiation Therapy, Image Technology, and Chemotherapy, and the Survival of Patients with

Nasopharyngeal Carcinoma: A Cohort Study From 1990 to 2012. International journal of radiation oncology, biology, physics 2019;105:581-90.

11. Tang LQ, Chen DP, Guo L, et al. Concurrent chemoradiotherapy with nedaplatin versus cisplatin in stage II-IVB nasopharyngeal carcinoma: an open-label, non-inferiority, randomised phase 3 trial. The Lancet Oncology 2018; 19:461-73.
12. Rotolo F, Pignon JP, Bourhis J, et al. Surrogate End Points for Overall Survival in Locoregionally Advanced Nasopharyngeal Carcinoma: An Individual Patient Data Meta-analysis. Journal of the National Cancer Institute 2017;109.
13. Li WZ, Lv X, Hu D, et al. Effect of Induction Chemotherapy with Paclitaxel, Cisplatin, and Capecitabine vs Cisplatin and Fluorouracil on Failure-Free Survival for Patients with Stage IVA to IVB Nasopharyngeal Carcinoma: A Multicenter Phase 3 Randomized Clinical Trial. JAMA oncology 2022; 8:706-14.
14. Ma J, Sun Y, Liu X, et al. PD-1 blockade with sintilimab plus induction chemotherapy and concurrent chemoradiotherapy (IC-CCRT) versus IC-CCRT in locoregionally-advanced nasopharyngeal carcinoma (LANPC): A multicenter, phase 3, randomized controlled trial (CONTINUUM). Journal of Clinical Oncology 2023;41: LBA6002-LBA.
15. Liu LT, Liu H, Huang Y, et al. Concurrent chemoradiotherapy followed by adjuvant cisplatin-gemcitabine versus cisplatin-fluorouracil chemotherapy for N2-3 nasopharyngeal carcinoma: a multicentre, open-label, randomised, controlled, phase 3 trial. The Lancet Oncology 2023.

Comment 4. Treatment procedure and compliance: When reviewing the treatment procedures, there were differences between the present trial and the prior induction chemotherapy trial of triplet versus doublet chemotherapy by the same group (Wang Zhong Li, et al. JAMA Oncol, 2022). Specifically, here, they used 3 cycles of TPX, while in the preceding trial, the investigators used 2 cycles of TPX, and here, they replaced taxol with abraxane. Additionally, they incorporated apatinib to TPX during the induction phase, and camrelizumab from induction throughout adjuvant phases of treatment; this would also imply that they have now investigated a 5-drug regimen for induction chemotherapy for high-risk NPC.

a. How were these patients staged? Was it N3 by 7th or 8th edition? This was not mentioned in the main text.

b. With this regimen, the authors reported a 26.5% complete response rate post-induction, but I am unsure about the tolerability of this 5-drug combo (see Comment #7).

c. What was the time to starting adjuvant camrelizumab? Was there any dose reduction or treatment interruption? Did adjuvant camrelizumab delay recovery from radiotherapy toxicities?

d. Perhaps more importantly, what was the scientific rationale for this quintet combination? Apatinib, which is a VEGFR2 small molecule inhibitor, has single agent activity in recurrent-metastatic NPC (Luo, et al. Oral Oncology, 2021). However, was there phase 1b data supporting the safety of this combination? This information was lacking in the Introduction and reporting of the results.

Response: Thank you very much for the comments.

For point 4 a:

Patients' staging was defined based on the criteria of AJCC 8th edition. We have added this statement

in revised manuscript that “Eligible patients were aged 18-65 years with previously untreated, histologically proven non-keratinising stage T_{any}N3M0 NPC (American Joint Committee on Cancer, 8th edition).”.

For point 4 b:

For the tolerability of this 5-drug regimen, the cumulative TRAE rate of 65% is similar to other studies (ranging from 58% to 76%; 75.7% in Zhang Y, et al. *N Engl J Med* 2019, 73% in Sun Y, et al. *Lancet Oncology* 2016, 65.8% and 57.6% in Li WZ, et al. *JAMA Oncology* 2022, 66.3% in Cao SM, et al. *Eur J Cancer* 2017, 75.3% in Ma J, et al. 2023 ASCO abstract LBA6002). Previous study showed that the proportion of patients completing 3 cycles of induction chemotherapy and 2 cycles of concurrent chemoradiotherapy ranges from 88% to 97% and 88% to 92%, respectively (Zhang Y, et al. *N Engl J Med* 2019, Sun Y, et al. *Lancet Oncology* 2016). In the present study, 100% (49/49) and 87% (41/47) patients received 3 cycles of induction chemotherapy and 2 cycles of concurrent chemoradiotherapy, respectively. This suggests that the combination regimen is well tolerated even with the addition of immunotherapy and anti-VEGFR2 agent.

Reference:

1. Zhang Y, Chen L, Hu GQ, et al. Gemcitabine and Cisplatin Induction Chemotherapy in Nasopharyngeal Carcinoma. *The New England journal of medicine* 2019; 381:1124-35.
2. Sun Y, Li WF, Chen NY, et al. Induction chemotherapy plus concurrent chemoradiotherapy versus concurrent chemoradiotherapy alone in locoregionally advanced nasopharyngeal carcinoma: a phase 3, multicentre, randomised controlled trial. *The Lancet Oncology* 2016; 17:1509-20.
3. Cao SM, Yang Q, Guo L, et al. Neoadjuvant chemotherapy followed by concurrent chemoradiotherapy versus concurrent chemoradiotherapy alone in locoregionally advanced nasopharyngeal carcinoma: A phase III multicentre randomised controlled trial. *European journal of cancer (Oxford, England: 1990)* 2017;75:14-23.
4. Li WZ, Lv X, Hu D, et al. Effect of Induction Chemotherapy with Paclitaxel, Cisplatin, and Capecitabine vs Cisplatin and Fluorouracil on Failure-Free Survival for Patients with Stage IVA to IVB Nasopharyngeal Carcinoma: A Multicenter Phase 3 Randomized Clinical Trial. *JAMA oncology* 2022; 8:706-14.
5. Ma J, Sun Y, Liu X, et al. PD-1 blockade with sintilimab plus induction chemotherapy and concurrent chemoradiotherapy (IC-CCRT) versus IC-CCRT in locoregionally-advanced nasopharyngeal carcinoma (LANPC): A multicenter, phase 3, randomized controlled trial (CONTINUUM). *Journal of Clinical Oncology* 2023;41: LBA6002-LBA.

For point 4 c:

As reported in the manuscript, camrelizumab is administrated intravenously at a fixed dose of 200 mg every 3 weeks from the first cycle of induction chemotherapy until 12 months. The camrelizumab is administered on a continuous course every three weeks. And, as protocol preset, fixed dose of camrelizumab was administrated except those weighing less than 40 kg, who received an actual dose of 3 mg/kg. In the trial protocol, we have set the detailed information of the interruption of camrelizumab. (Protocol, Second version, In **6.3.5 Camrelizumab** section). And the compliance data of the camrelizumab was reported in **Supplement Table S5**.

Due to the nature of the single arm study, it's unclear if camrelizumab maintenance therapy delays

recovery from radiotherapy toxicities, but we observed no increased in radiotherapy toxicities rates compared with historical data. In this study, in the concurrent phase, 3-4 TRAE rate of 44% is similar to the other studies (48% in *Cao SM, et al. Eur J Cancer 2017*, and 54% in *Sun Y, et al. Lancet Oncology 2016*). Meanwhile, based on the results of abstract LBA6002 study in recent 2023 ASCO annual meeting, compared with chemoradiotherapy, anti-PD-1 agents combined with chemoradiotherapy obtain similar quality of life and radiotherapy discontinuation rate(0.5% vs. 0.95%, respectively) compared to standard-of-care group (*Ma J, et al. 2023 ASCO abstract LBA6002*).

Reference:

1. Cao SM, Yang Q, Guo L, et al. Neoadjuvant chemotherapy followed by concurrent chemoradiotherapy versus concurrent chemoradiotherapy alone in locoregionally advanced nasopharyngeal carcinoma: A phase III multicentre randomised controlled trial. *European journal of cancer (Oxford, England : 1990)* 2017;75:14-23.
2. Sun Y, Li WF, Chen NY, et al. Induction chemotherapy plus concurrent chemoradiotherapy versus concurrent chemoradiotherapy alone in locoregionally advanced nasopharyngeal carcinoma: a phase 3, multicentre, randomised controlled trial. *The Lancet Oncology* 2016; 17:1509-20.
3. Ma J, Sun Y, Liu X, et al. PD-1 blockade with sintilimab plus induction chemotherapy and concurrent chemoradiotherapy (IC-CCRT) versus IC-CCRT in locoregionally-advanced nasopharyngeal carcinoma (LANPC): A multicenter, phase 3, randomized controlled trial (CONTINUUM). *Journal of Clinical Oncology* 2023;41: LBA6002-LBA.

For point 4 d:

We acknowledge that angiogenesis is a validated target in NPC, and several clinical trials have reported that antiangiogenic therapy improved the efficacy of immunotherapy. Indeed, the combination of antiangiogenic therapy with immunotherapy has been approved by the US Food and Drug Administration for the treatment of endometrial carcinoma, hepatocellular carcinoma, and renal carcinoma (*Makker V, et al. N Engl J Med 2022; Finn RS, et al. N Engl J Med 2020; Motzer RJ, et al. N Engl J Med 2019; Rini BI, et al. N Engl J Med 2019*). Nevertheless, monotherapy of apatinib has showed activity against NPC in several reports (*Huang L, et al. Oral Oncology 2021, Li L, et al. Invest New Drugs 2020, Zhou L, et al. Drug Des Devel Ther 2020*). Furthermore, apatinib synergizes antitumoral effects of camrelizumab in several solid tumors, including advanced NPC, hepatocellular carcinoma, breast cancer, and osteosarcoma (*Ding X, et al. Journal of Clinical Oncology 2023, Xu J, et al. Clinical Cancer Research 2019, Li Q, et al. Clinical Cancer Research 2020, Liu J, et al. J Immunother Cancer 2020, Xie L, et al. J Immunother Cancer 2020*). And recently, apatinib plus immunotherapy (and chemotherapy) demonstrated promising antitumor activity and manageable toxicity in patients with recurrent/metastatic NPC (*Ding X, et al. Journal of Clinical Oncology 2023, You R, et al. Med 2022*). We have revised the description in the portion of Introduction as the following “*Meanwhile, antiangiogenic agent apatinib, an orally administered vascular endothelial growth factor receptor 2 (VEGFR2) tyrosine kinase inhibitor, has shown antitumoral activity and synergetic effects with camrelizumab in several solid tumors, including advanced NPC, hepatocellular carcinoma, breast cancer, and osteosarcoma.*”.

Reference:

1. Makker V, Colombo N, Casado Herraes A, et al: Lenvatinib plus pembrolizumab for advanced endometrial cancer. *New Engl J Med* 386:437-448, 2022.
2. Finn RS, Qin S, Ikeda M, et al: Atezolizumab plus bevacizumab in unresectable hepatocellular carcinoma. *New Engl J Med* 382:1894-1905, 2020.
3. Motzer RJ, Penkov K, Haanen J, et al: Avelumab plus axitinib versus sunitinib for advanced renal-cell carcinoma. *New Engl J Med* 380:1103-1115, 2019.
4. Rini BI, Plimack ER, Stus V, et al: Pembrolizumab plus axitinib versus sunitinib for advanced renal-cell carcinoma. *New Engl J Med* 380:1116-1127, 2019.
5. Huang L, Zhang X, Bai Y, et al. Efficacy and safety of apatinib in recurrent/metastatic nasopharyngeal carcinoma: A pilot study. *Oral oncology* 2021; 115:105222.
6. Li L, Kong F, Zhang L, et al. Apatinib, a novel VEGFR-2 tyrosine kinase inhibitor, for relapsed and refractory nasopharyngeal carcinoma: data from an open-label, single-arm, exploratory study. *Investigational new drugs* 2020; 38:1847-53.
7. Ding X, Zhang WJ, You R, et al. Camrelizumab Plus Apatinib in Patients with Recurrent or Metastatic Nasopharyngeal Carcinoma: An Open-Label, Single-Arm, Phase II Study. *Journal of clinical oncology: official journal of the American Society of Clinical Oncology* 2023; 41:2571-82.
8. Xu J, Zhang Y, Jia R, et al. Anti-PD-1 Antibody SHR-1210 Combined with Apatinib for Advanced Hepatocellular Carcinoma, Gastric, or Esophagogastric Junction Cancer: An Open-label, Dose Escalation and Expansion Study. *Clinical cancer research: an official journal of the American Association for Cancer Research* 2019; 25:515-23.
9. Li Q, Wang Y, Jia W, et al. Low-Dose Anti-Angiogenic Therapy Sensitizes Breast Cancer to PD-1 Blockade. *Clinical cancer research: an official journal of the American Association for Cancer Research* 2020; 26:1712-24.
10. Liu J, Liu Q, Li Y, et al. Efficacy and safety of camrelizumab combined with apatinib in advanced triple-negative breast cancer: an open-label phase II trial. *J Immunother Cancer* 2020;8.
11. Xie L, Xu J, Sun X, et al. Apatinib plus camrelizumab (anti-PD1 therapy, SHR-1210) for advanced osteosarcoma (APFAO) progressing after chemotherapy: a single-arm, open-label, phase 2 trial. *J Immunother Cancer* 2020;8.
12. You R, Zou X, Ding X, et al. Gemcitabine combined with apatinib and toripalimab in recurrent or metastatic nasopharyngeal carcinoma. *Med* 2022; 3:664-81 e6.

Comment 5. Results (DMFS):

a. Choice of DMFS as the primary endpoint: This reviewer queries the choice of this endpoint at 1-y mark, especially when the entire treatment duration was longer than 1 year.

b. How did they estimate the time to event? Was it from the start of treatment? If so, then this reviewer would contest that the choice of a 1-y endpoint is flawed, since it is highly unlikely for patients to progress within such a short window. It would be difficult to judge if this regimen was indeed “effective” in eradicating the occult distant metastases. At least, the authors reported on the 2-year survival rates.

Response: Thanks for your critical comments. As *response to Reviewer #1 Comment #3* mentioned, distant metastasis is the most important determinant of the survival rates for N3 patients despite the availability of aggressive treatment strategies of chemotherapy and radiation therapy, and about half of distant metastasis events occurred within 1 year (*Sun X, et al. Radiology and Oncology 2014*). That is the main reason that we set up primary endpoint at 1-year mark. In addition, given economic and time considerations, 1-year mark setting is logical and legitimate in this phase 2 trial. In addition, 2- and 3-year DMFS were pre-set as secondary endpoints in the present study. DMFS was calculated from the date of treatment initiation to the date of documented distant metastasis. With the median follow-up duration of 28.7 months (ranged from 22.7 to 36.7), the 98% (47/48 alive) patients were followed up for over 24 months. In our manuscript, we also reported 2-year endpoints which is sufficiently representative of long-term efficacy as indicated based on previous reports (*Tang LQ, et al. Lancet Oncology 2018; Rotolo F, et al. J Natl Cancer Inst 2017*). Results showed that the present QUINTUPLED regimen has promising disease control in N3 NPC patients. (**Supplementary Figure S6** in revised Appendix and **Figure 1-2 of Reviewer #1 Comment #3**; *Li WZ, et al. JAMA Oncology 2022; Ma J, et al. 2023 ASCO abstract LBA6002; Liu LT, et al. Lancet Oncology 2023*).

Reference:

1. Sun X, Su S, Chen C, et al. Long-term outcomes of intensity-modulated radiotherapy for 868 patients with nasopharyngeal carcinoma: an analysis of survival and treatment toxicities. *Radiation therapy and oncology: journal of the European Society for Therapeutic Radiology and Oncology* 2014; 110:398-403.
2. Chen YP, Liu X, Zhou Q, et al. Metronomic capecitabine as adjuvant therapy in locoregionally advanced nasopharyngeal carcinoma: a multicentre, open-label, parallel-group, randomised, controlled, phase 3 trial. *Lancet (London, England)* 2021;398:303-13.
3. Tang LQ, Chen DP, Guo L, et al. Concurrent chemoradiotherapy with nedaplatin versus cisplatin in stage II-IVB nasopharyngeal carcinoma: an open-label, non-inferiority, randomised phase 3 trial. *The Lancet Oncology* 2018; 19:461-73.
4. Rotolo F, Pignon JP, Bourhis J, et al. Surrogate End Points for Overall Survival in Loco-Regionally Advanced Nasopharyngeal Carcinoma: An Individual Patient Data Meta-analysis. *Journal of the National Cancer Institute* 2017;109.
5. Li WZ, Lv X, Hu D, et al. Effect of Induction Chemotherapy with Paclitaxel, Cisplatin, and Capecitabine vs Cisplatin and Fluorouracil on Failure-Free Survival for Patients with Stage IVA to IVB Nasopharyngeal Carcinoma: A Multicenter Phase 3 Randomized Clinical Trial. *JAMA oncology* 2022; 8:706-14.
6. Ma J, Sun Y, Liu X, et al. PD-1 blockade with sintilimab plus induction chemotherapy and concurrent chemoradiotherapy (IC-CCRT) versus IC-CCRT in locoregionally-advanced nasopharyngeal carcinoma (LANPC): A multicenter, phase 3, randomized controlled trial (CONTINUUM). *Journal of Clinical Oncology* 2023;41: LBA6002-LBA.
7. Liu LT, Liu H, Huang Y, et al. Concurrent chemoradiotherapy followed by adjuvant cisplatin-gemcitabine versus cisplatin-fluorouracil chemotherapy for N2-3 nasopharyngeal carcinoma: a multicentre, open-label, randomised, controlled, phase 3 trial. *The Lancet Oncology* 2023.

Comment 6. Results (Response to induction treatment): I have the following comments on the evaluation of response to induction treatment.

a. Foremost, there seems to be a discordance between radiological response and pathological response (73.5% PR by RECIST but 84.8% pathological -ve biopsy). Could this be due to a sampling error?

b. This reviewer is also intrigued by the modest CR rate of 26.5%, even with a quintet combination. With such an intensive regimen, one might have expected a higher CR rate. To interpret the data better, more information ought to be included in Table 1. For example, were the nodes bulky? How many levels were involved? What was the median gross tumour volume for the primary and nodal lesions? See my Comment #4a on stage classification.

c. Perhaps, it would have been very useful to have EBV DNA response data to the induction phase. What was the proportion of patients who had a complete clearance by the end of induction treatment? What was the magnitude of drop in EBV DNA counts?

d. Overall, this reviewer is unconvinced that the pathological responses are indicative of efficacy to this quintet combo. Moreover, conventional imaging, unless PET-CT, is not ideal for assessing CR of nodes; as a case in point, a lymph node of 3 cm size shrinking to 0.8 cm would be considered as PR? However, it could well be a pathological CR, since involved lymph nodes don't typically shrink entirely even months post-treatment (Li WF, et al. Oral Oncology, 2017). If anything, EBV DNA and PET post-induction would have been highly informative of efficacy, especially since the concern with patients with N3 NPC is the risk of distant metastases - thus, the current measures do not offer insights in this regard.

e. Perhaps, the authors could include some pre- and post-induction images of extreme responders to the induction treatment, which would be informative to the readers.

Response: We thank the reviewer for the constructive comments.

For point 6 a:

We agree with the chance of sampling error in this study. Biopsy is not mandatory for patients who were enrolled to this trial and only 33 patients who consented to be biopsied. However, physicians who performed this kind of biopsy procedure are experienced in one of largest clinical center of NPC treatment in the world, and we have tried to minimize this kind of error.

For point 6 b:

We have added the additional key information, as the following shown, in **Table 1**.

Lymph node levels	
Retropharyngeal	48
I	9
II	49
III	49
IV	40
Va	28

Vb	18
> 6 cm	13

For point 6 c:

We did provide the EBV DNA response data in supplemental Table 7. We are showing this set of data here again to address the reviewer’s concern. As we show as following, there are 48 (98%) patients who had positive plasma EBV DNA level at baseline. The EBV DNA levels became persistently undetectable in 46/48 (96%) of patients. The 41/48 (85%) and 5/48 (10%) of patients whose EBV DNA levels changed to undetectable by the end of induction therapy phase and radiotherapy phase, respectively. The remaining two patients had persistently detectable plasma EBV DNA, including one with regional PR and another suffering death due to the disease progression.

Supplemental Table S7. Plasma EBV DNA clearance information.*

	No. (%) of patients
No. patients evaluated	49 (100%)
Patients who had positive plasma EBV DNA level at baseline	48 (98%)
Persistently detectable	2 (4%)
Change to undetectable	46 (96%)
Median (IQR) clearance time, days	36 (27-56)
Range, days	7-106
Undetectable during induction therapy phase	41 (85%)
After 1 st cycle of induction therapy	15 (31%)
After 2 nd cycle of induction therapy	17 (35%)
After 3 rd cycle of induction therapy	9 (19%)
Undetectable during radiotherapy phase	5 (10%)

IQR = interquartile range.

* Quantification testing of plasma Epstein-Barr virus (EBV) DNA level were based on a lower limit of detection cutoff value of 1000 copies/mL. Here the undetectable level meaning whose EBV DNA level is reported with 0 copies/mL.

For point 6 d:

As response to Reviewer #1 Comment #6 c mentioned, the EBV DNA response data have been reported in **Supplemental Table S7**. And the longitudinal monitoring trajectories of the change in plasma EBV DNA concentrations was given in **Supplemental Figure S4**. Taking into account the practicalities and financial implications in real-world practice, PET/CT is not routinely recommended for tumor staging and evaluation. Based on previous high-quality studies (*Zhang Y, et al. N Engl J Med 2019, Sun Y, et al. Lancet Oncology 2016*), nasopharyngeal and cervical MR is used for examining the nasopharynx and cervical lymph nodes in this study. Furthermore, tumor response, as only one of the secondary endpoints, was assessed by the RECIST version 1.1 (*Eisenhauer EA, et al. Eur J Cancer 2009*). According to RECIST version 1.1 guideline, the same method of assessment and the same technique should be used to characterize each identified and reported lesion at baseline and during follow-up.

Reference:

1. Eisenhauer EA, Therasse P, Bogaerts J, et al. New response evaluation criteria in solid tumours: revised RECIST guideline (version 1.1). *European journal of cancer (Oxford, England: 1990)* 2009;45:228-47.
2. Zhang Y, Chen L, Hu GQ, et al. Gemcitabine and Cisplatin Induction Chemotherapy in Nasopharyngeal Carcinoma. *The New England journal of medicine* 2019; 381:1124-35.
3. Sun Y, Li WF, Chen NY, et al. Induction chemotherapy plus concurrent chemoradiotherapy versus concurrent chemoradiotherapy alone in locoregionally advanced nasopharyngeal carcinoma: a phase 3, multicentre, randomised controlled trial. *The Lancet Oncology* 2016; 17:1509-20.

For point 6 e:

We did provide representative pre- and post-induction radiological images in **Supplemental Figure S2, and histological results in Supplemental Figure S3**, as the following Figures showed:

Supplemental Figure S2. Representative MRI comparison of pre- and after induction phase. Panel A (Right cavernous sinus invasion), B (Right oropharyngeal invasion), C (Right cavernous sinus, clivus, petrous apex, and area II-IV lymph node invasion) and D (Left cavernous sinus, retropharyngeal and area II-V lymph nodes invasion) denote nasopharyngeal and neck MRI before treatment. Panel E, F, G, and H denote complete response to induction therapy in the corresponding aforementioned sites, respectively.

Supplemental Figure S3. Pathology of nasopharyngeal and lymph node tissues. Panel A and E denote hematoxylin-eosin staining of nasopharyngeal tissue and lymph node tissue before treatment. Panel B and F denote immunohistochemistry staining of cell keratins (CK, positive) in nasopharyngeal tissue and lymph node tissue before treatment, respectively. Panel C and G denote situ hybridization of Epstein-Barr encoding region (EBER, positive) in nasopharyngeal tissue and lymph node tissue before treatment, respectively. Panel D and H denote hematoxylin-eosin staining

of nasopharyngeal tissue and lymph node tissue after induction therapy showing pathological complete response, respectively.

Comment 7. Results - Figure 2: While the data compared with the historical cohort looks promising, such an analysis is flawed. Was the historical cohort matched in some way? Otherwise, this reviewer will urge the investigators to remove the historical cohort as its inclusion will lead to misinterpretation.

Response: Thanks for your suggestive comment. No matching was done. The characteristics of the patients at baseline between QUINTUPLED and historical cohorts were given in Supplemental Table S3. We have removed the curve of historical cohort from Figure 2 as suggested. In addition, the original Figure 2 was placed in the Supplementary Material as **Figure S6, in case** some readers may be interested in the comparison of long-term efficacy from different treatment cohorts.

Comment 8. Toxicity results: The incidence of 65.3% (32 of 49) patients for G3-4 treatment-related adverse events is substantial. This reviewer would not consider such a regimen to be “well-tolerated”. Moreover, the incidence of 46.9% (23 of 39) patients with G3-4 adverse events during induction chemotherapy is substantial, and this reviewer is surprised that dose intensities for the chemotherapeutic agents remained at 100%?

Response: Thank you for your comments. Although the present combination regimen is a more aggressive therapeutic strategy, it does not result in any unexpected grade 3 or 4 TRAEs or irAEs. The cumulative TRAE rate of 65% is similar to other studies (ranging from 58% to 76%) (75.7% in Zhang Y, et al. *N Engl J Med* 2019, 73% in Sun Y, et al. *Lancet Oncology* 2016, 65.8% and 57.6% in Li WZ, et al. *JAMA Oncology* 2022, 66.3% in Cao SM, et al. *Eur J Cancer* 2017, 75.3% in Ma J, et al. 2023 ASCO abstract LBA6002). Previous study showed that the proportion of patients completing 3 cycles of induction chemotherapy and 2 cycles of concurrent chemoradiotherapy ranged from 88% to 97% and 88% to 92%, respectively (Zhang Y, et al. *N Engl J Med* 2019, Sun Y, et al. *Lancet Oncology* 2016). In the present study, 100% (49/49) and 87% (41/47) patients received 3 cycles of induction chemotherapy and 2 cycles of concurrent chemoradiotherapy, respectively. This suggests that the QUINTUPLED regimen is well tolerated even with the addition of immunotherapy and anti-VEGFR2 agent. In addition, we have provided objective information about compliance of chemotherapeutic agents rather than description of the median dose intensities in order to avoid confusion (**Supplemental Table S4**).

Reference:

1. Li WZ, Lv X, Hu D, et al. Effect of Induction Chemotherapy with Paclitaxel, Cisplatin, and Capecitabine vs Cisplatin and Fluorouracil on Failure-Free Survival for Patients with Stage IVA to IVB Nasopharyngeal Carcinoma: A Multicenter Phase 3 Randomized Clinical Trial. *JAMA oncology* 2022; 8:706-14.
2. Sun Y, Li WF, Chen NY, et al. Induction chemotherapy plus concurrent chemoradiotherapy versus concurrent chemoradiotherapy alone in locoregionally advanced nasopharyngeal carcinoma: a phase 3, multicentre, randomised controlled trial. *The Lancet Oncology* 2016; 17:1509-20.
3. Zhang Y, Chen L, Hu GQ, et al. Gemcitabine and Cisplatin Induction Chemotherapy in Nasopharyngeal Carcinoma. *The New England journal of medicine* 2019; 381:1124-35.
4. Cao SM, Yang Q, Guo L, et al. Neoadjuvant chemotherapy followed by concurrent chemoradiotherapy versus concurrent chemoradiotherapy alone in locoregionally advanced nasopharyngeal carcinoma: A phase III multicentre randomised controlled trial. *European journal of cancer (Oxford, England: 1990)* 2017;75:14-23.
5. Ma J, Sun Y, Liu X, et al. PD-1 blockade with sintilimab plus induction chemotherapy and concurrent chemoradiotherapy (IC-CCRT) versus IC-CCRT in locoregionally-advanced

nasopharyngeal carcinoma (LANPC): A multicenter, phase 3, randomized controlled trial (CONTINUUM). Journal of Clinical Oncology 2023;41: LBA6002-LBA.

Comment 9. Discussion: There need to be some elaboration on the mechanisms underpinning the efficacy of this regimen in targeting occult metastases in NPC.

Response: We thank for this constructive comment. It is largely unknown regarding the exact mechanism on controlling distant metastasis as well as occult metastasis with our regimen. However, there are some known mechanisms which may contribute to this superior outcome. Emerging evidence has demonstrated that the antiangiogenic agents not only normalize tumor vascular network, but modulate tumor immune environment, including promoting T-cell infiltration, inducing M1 macrophage polarization, decreasing the recruitment of Treg and myeloid suppressor cell, and downregulating inhibitory immune checkpoints such as PD-L1, thereby it converts the tumor immune environment from immunosuppressive to immune-active (Zhao S, et al. Cancer Immunol Res 2019; Huang Y, et al. Proc Natl Acad Sci 2012; Du Four S, et al. Am J Cancer Res 2016; Jain RK, et al. Cancer Cell 2014). These sets of evidence suggest that combining antiangiogenic agents with immunotherapy improves efficacy by creating positive feedback loops that reinforce each other. Meanwhile, the immunostimulatory effects of radiation has also been demonstrated (Arina A, et al. Clinical cancer research 2020; Galluzzi L, et al. Nature reviews Clinical oncology 2023; Salas-Benito D, et al. Cancer Discovery 2021.). We are working on single-cell sequencing analysis derived from pre- and post-treatment biopsied samples, which may help us to identify the potential involved mechanism.

Reference:

1. Zhao S, Ren S, Jiang T, et al: Low-Dose Apatinib Optimizes Tumor Microenvironment and Potentiates Antitumor Effect of PD-1/PD-L1 Blockade in Lung Cancer. Cancer Immunol Res 7:630-643, 2019.
2. Huang Y, Yuan J, Righi E, et al: Vascular normalizing doses of antiangiogenic treatment reprogram the immunosuppressive tumor microenvironment and enhance immunotherapy. Proc Natl Acad Sci U S A 109:17561-6, 2012.
3. Du Four S, Maenhout SK, Niclou SP, et al: Combined VEGFR and CTLA-4 blockade increases the antigen-presenting function of intratumoral DCs and reduces the suppressive capacity of intratumoral MDSCs. Am J Cancer Res 6:2514-2531, 2016.
4. Jain RK: Antiangiogenesis strategies revisited: from starving tumors to alleviating hypoxia. Cancer Cell 26:605-22, 2014.
5. Arina A, Gutiontov SI, Weichselbaum RR. Radiotherapy and Immunotherapy for Cancer: From "Systemic" to "Multisite". Clinical cancer research: an official journal of the American Association for Cancer Research 2020; 26:2777-82.
6. Galluzzi L, Aryankalayil MJ, Coleman CN, Formenti SC. Emerging evidence for adapting radiotherapy to immunotherapy. Nature reviews Clinical oncology 2023.
7. Salas-Benito D, Perez-Gracia JL, Ponz-Sarvisé M, et al. Paradigms on Immunotherapy Combinations with Chemotherapy. Cancer discovery 2021; 11:1353-67.

Responses to Reviewer #2

Comment 1. Needs editing to improve English fluency.

Response: Thanks for your comments. We have revised the manuscript with the help from a native English speaker and professional scientist Dr. Firouzeh Korangy.

Comment 2. *Why have the investigators selected a relatively obscure IC regimen to build on? Most use either GC or TPF lite. This makes interpretation of these results a bit challenging.*

Response: We appreciate reviewer's expertise input. Indeed, our recent published phase 3 randomized clinical trial has shown that induction chemotherapy with the regimen of TPC (paclitaxel, cisplatin and capecitabine) is more efficacious than that with PF (cisplatin and CI 5-FU) for patients with stage IVA to IVB NPC (*Li WZ, et al. JAMA Oncology 2022*). Currently, the TPC regimen has become Level I recommendation with Class IA evidence in the Chinese Society of Clinical Oncology (CSCO) guideline. However, our study also demonstrated that none of the N3 patients reached CR status treated with TPC regimen, and 2-year PFS rate with TPC regimen was only 85%. Therefore, we assume that the addition of the 2 new biological drugs may improve the therapeutic efficacy based on their individual mechanism of action and evidence of synergistic antitumoral effect when used with the combination of these two new drugs. Meanwhile, based on our previous trial and clinical experience of paclitaxel, the incidence of its allergy was 9% (*Li WZ, et al. JAMA Oncology 2022*), while no allergic events occurred in NAB-paclitaxel (*Ke LR, et al. Oral oncology 2017*). In addition, our recent phase 2 trial has shown encouraging anti-tumor effects and manageable toxicities in locally advanced NPC using NAB-paclitaxel (*Ke LR, et al. Oral oncology 2017*). In order to decrease dose-limiting toxicities of paclitaxel but improve antitumoral efficacy via increasing intratumoral concentration of paclitaxel, we replaced the paclitaxel with NAB-paclitaxel in the present trial. It is worth to mention that we are finishing a clinical trial initiated in 2019 to compare the efficacy of NAB-TPC versus GP in advanced NPC. The upcoming public data shows that NAB-TPC regimen provided superior PFS versus GP regimen for advanced NPC at the first-line setting (Median IRC-assessed PFS, 11.3 m versus 7.7 m, p=0.002; chicttr.org.cn registration number: ChiCTR1900027112). It is impressive that the QUINTUPLED regimen obtained a high CR rate of 26% after induction phase. We acknowledge that it would be hard to tease out the real factors which contributed to higher response rate because of the nature of the trial design. However, in our previously reported phase 3 clinical trial (*Li WZ, et al. JAMA Oncology 2022*), we did observe a much-lower but similar CR rate to induction chemotherapy between TPC and PF regimen (1.7% and 3.3%, respectively), which indicates that the higher CR rate would derive from the addition of the two new drugs rather than the switch from CI 5FU to capecitabine. At the same time, in terms of the switch from CI 5FU to capecitabine, similar tumor response rates were observed between 5-FU and capecitabine groups in both anal cancer and advanced metastatic gastric cancer (*Jones CM, et al, Int J Radiat Oncol Biol Phys 2018; Chen, J et al, Asia Pac J Clin Oncol 2018*), as well as in rectal cancer (*O'Connell MJ, et al, J Clin Oncol 2014*). Therefore, the addition of the 2 new drugs would be the main contributor for the higher response rate.

Reference:

1. Li WZ, Lv X, Hu D, et al. Effect of Induction Chemotherapy with Paclitaxel, Cisplatin, and Capecitabine vs Cisplatin and Fluorouracil on Failure-Free Survival for Patients with Stage IVA to IVB Nasopharyngeal Carcinoma: A Multicenter Phase 3 Randomized Clinical Trial. *JAMA oncology* 2022; 8:706-14.
2. Ke LR, Xia WX, Qiu WZ, et al. A phase II trial of induction NAB-paclitaxel and cisplatin

followed by concurrent chemoradiotherapy in patients with locally advanced nasopharyngeal carcinoma. *Oral oncology* 2017; 70:7-13.

3. Jones CM, Adams R, Downing A, et al. Toxicity, Tolerability, and Compliance of Concurrent Capecitabine or 5-Fluorouracil in Radical Management of Anal Cancer with Single-dose Mitomycin-C and Intensity Modulated Radiation Therapy: Evaluation of a National Cohort. *International journal of radiation oncology, biology, physics* 2018;101:1202-11.
4. Chen J, Xiong J, Wang J, Zheng L, Gao Y, Guan Z. Capecitabine/cisplatin versus 5-fluorouracil/cisplatin in Chinese patients with advanced and metastatic gastric cancer: Re-analysis of efficacy and safety data from the ML17032 phase III clinical trial. *Asia-Pacific journal of clinical oncology* 2018; 14: e310-e6.
5. O'Connell MJ, Colangelo LH, Beart RW, et al. Capecitabine and oxaliplatin in the preoperative multimodality treatment of rectal cancer: surgical end points from National Surgical Adjuvant Breast and Bowel Project trial R-04. *Journal of clinical oncology: official journal of the American Society of Clinical Oncology* 2014; 32:1927-34.

Comment 3. *It is unclear that data were behind the modification during the study of apatinib dosing from continuous to just 5 d/ week.*

Response: Thanks for your comments. The usage of apatinib has been modified during the study. At the beginning of the study, the schedule and dosage of apatinib was designed as day 1 to 56, 250 mg/day. In the initial stages of enrollment, we found that more than half of patients (17/25, 68%) were unable to tolerate continuous oral dosing and experienced discontinuation (detailed information of compliance to apatinib in Supplement Table S5). Due to the tolerability of the enrolled patients, we modified the initial dosage of apatinib in protocol, second version. The change was reported in the Summary of changes in protocol. As a result, after modifying the initial dosage of apatinib, more patients were able to tolerate compared to original dosing (p value = 0.032), and only 9/24 (37.5%) patients with dose interruptions. The compliance data of the apatinib was reported in **Supplement Table S5**.

Protocol version	Usage of apatinib	Maximum dose	Number of patients received
First	Apatinib 250 mg is recommend orally administrated daily, 7 days every week.	14000 mg	25
Second (Final)	Apatinib taken orally at a dose of 250 mg daily on days 1 to 5 per week for a total of 8 weeks.	10000 mg	24

Comment 4. *It is surprising that tumor response assessment did not include PET CT nor ct EBV DNA. PET CT in particular is SOC many places for the evaluation 12-16 weeks post XRT and having only CT and MRI could lead to significant underdetection of persistent or recurrent disease.*

Response: Thanks for your comments. Tumor response assessment is evaluated according to the RECIST, version 1.1. After radiation therapy finished, follow-up evaluations were performed every 3 months until 3 years, every 6 months until 5 years, and annually thereafter. Although PET/CT has ideal specificity and sensitivity for the diagnosis of distant metastases, the current high cost of

PET/CT limits its wide application in the follow-up of NPC. In our study, PET CT is optimal but not mandatory. As mentioned in the CSCO guideline, currently, chest and abdomen CT and whole-body bone imaging are used commonly in the routine follow-up of NPC. Indeed, the follow up strategy of our study compliant with CSCO guideline (Zhou GQ, et al. 2020 *Nature Communications*; Tang LL, et al. 2021 *Cancer Communications*). According to RECIST version 1.1 guideline, the same method of assessment and the same technique should be used to characterize each identified and reported lesion at baseline and during follow-up. As with previous high-quality studies, nasopharyngeal and cervical MR is the most commonly used and preferred test for examining the nasopharynx and cervical lymph nodes, taking into account the practical and financial standpoints (Zhang Y, et al. *N Engl J Med* 2019, Sun Y, et al. *Lancet Oncology* 2016). Meanwhile, plasma EBV DNA test were done at every 3-month follow-up timepoint (As shown in **Supplemental Figure 4. Trajectories of plasma EBV DNA levels**). And the plasma EBV DNA clearance information was also provided in **Supplemental Table S7**.

Reference:

1. Zhou GQ, Wu CF, Deng B, et al. An optimal posttreatment surveillance strategy for cancer survivors based on an individualized risk-based approach. *Nature communications* 2020; 11:3872.
2. Tang LL, Chen YP, Chen CB, et al. The Chinese Society of Clinical Oncology (CSCO) clinical guidelines for the diagnosis and treatment of nasopharyngeal carcinoma. *Cancer Commun (Lond)* 2021; 41:1195-227.
3. Eisenhauer EA, Therasse P, Bogaerts J, et al. New response evaluation criteria in solid tumours: revised RECIST guideline (version 1.1). *European journal of cancer (Oxford, England: 1990)* 2009;45:228-47.
4. Zhang Y, Chen L, Hu GQ, et al. Gemcitabine and Cisplatin Induction Chemotherapy in Nasopharyngeal Carcinoma. *The New England journal of medicine* 2019; 381:1124-35.
5. Sun Y, Li WF, Chen NY, et al. Induction chemotherapy plus concurrent chemoradiotherapy versus concurrent chemoradiotherapy alone in locoregionally advanced nasopharyngeal carcinoma: a phase 3, multicentre, randomised controlled trial. *The Lancet Oncology* 2016; 17:1509-20.

Comment 5. *Were PD-L1 TPS or CPS data collected on tumors?*

Response: Thanks for your comments. We did not screen patients according to PD-L1 status at baseline, because the majority of patients from the endemic region had PD-L1-positive tumors and the predictive value of PD-L1 expression in nasopharyngeal carcinoma patients undergoing immunotherapy is inconclusive (Lv JW, et al. 2019 *J Immunother Cancer*; Wang FH, et al. 2021 *JCO*).

Reference:

1. Lv JW, Li JY, Luo LN, Wang ZX, Chen YP. Comparative safety and efficacy of anti-PD-1 monotherapy, chemotherapy alone, and their combination therapy in advanced nasopharyngeal carcinoma: findings from recent advances in landmark trials. *J Immunother Cancer* 2019; 7:159.

2. Wang FH, Wei XL, Feng J, et al. Efficacy, Safety, and Correlative Biomarkers of Toripalimab in Previously Treated Recurrent or Metastatic Nasopharyngeal Carcinoma: A Phase II Clinical Trial (POLARIS-02). *Journal of clinical oncology: official journal of the American Society of Clinical Oncology* 2021; 39:704-12.

Comment 6. *It seems that the timing of biopsies on IC was not consistent. Some after one, some after two cycles, and only a fraction of the 49 patients had either biopsies.*

Response: Thanks for your comments. As the part of exploratory analysis and translational study in protocol, the purpose of lymph node puncture examination is to perform exploratory study with the goal of understanding potential mechanisms in addition to assessing pathological response. Biopsies should only be performed with the patient's consent and is not mandatory in this trial. There is no pre-set timing of biopsies on IC. During the initial stage of the study, we found that the enlarged lymph nodes had either disappeared or shrunk to the point that they were not suitable for biopsied after three courses of IC. As a result, we performed biopsies after two cycles of IC, that we can be sure to obtain residual tumors for the exploratory analysis of the tumor characteristics and underlying mechanisms of the treatment on tumor microenvironments. In addition, the pathologic results of biopsied samples after different courses were reported in Table 2.

Comment 7. *In the results, ctEBVDNA are reported. See item 4 above. the prespecified plan and ctEBV DNA collection and evaluation methods should be specified. Or was this as hoc? How is "persistently" negative ctEBV DNA defined?*

Response: Thanks for your comments. Plasma EBV DNA test were done at every routine follow-up timepoint (As shown in **Supplemental Figure 4. Trajectories of plasma EBV DNA levels**). As the part of prespecified plan, the ctEBV DNA collection and evaluation methods were given in **Supplement Appendix** (in the section of Plasma Epstein-Barr virus DNA extraction and real-time quantitative polymerase chain reaction analysis). "Persistently negative ctEBV DNA" is defined as a quantitative EBV DNA level reported as 0 copies/mL and also reported as zero on subsequent tests, and we have added the statement in corresponding Supplement Appendix (in the section of Plasma Epstein-Barr virus DNA extraction and real-time quantitative polymerase chain reaction analysis).

Comment 8. *Do the authors have a specific reason for having chosen camrelizumab for which reactive cutaneous capillary endothelial proliferation seems to be a challenge?*

Response: We appreciate reviewer's expertise input. Camrelizumab is a humanized immunoglobulin G4/k programmed cell death-1 (PD-1) monoclonal antibody, and previous study reported that camrelizumab is well tolerated with antitumor activity in NPC patients before the time of design of this study in 2019 (*Markham A, et al. Drugs 2019; Fang WF, et al. 2018 Lancet Oncol*). As reported in study, we acknowledge that the cumulative incidence of RCCEP in our study was higher than that reported in the CAPTAIN-1st study using combination regimen of camrelizumab plus GC to treat R/M NPC (86% versus 58%), but RCCEP was mild and controllable, and most went away spontaneously in a month after stopping the medication. And the main reasons of high incidence rate of RCCEP may result from the multiple-drug interactive effects and intensive toxicity monitoring (maintaining close communication with patients and their families) in our study on the setting of limited sample size. In addition, previous studies indicated that the occurrence of RCCEP is positively correlated with efficacy and could be a clinical biomarker for predicting efficacy of

camrelizumab (*Expert consensus in China 2020; Huang J, et al. Lancet Oncology 2020; Wang F, et al. Journal of hematology & oncology 2020*).

Reference:

1. Markham A, Keam SJ. Camrelizumab: First Global Approval. *Drugs* 2019; 79:1355-61.
2. Fang W, Yang Y, Ma Y, et al. Camrelizumab (SHR-1210) alone or in combination with gemcitabine plus cisplatin for nasopharyngeal carcinoma: results from two single-arm, phase 1 trials. *The Lancet Oncology* 2018; 19:1338-50.
3. Expert consensus on the clinical diagnosis and treatment of reactive cutaneous capillary endothelial proliferation caused by camrelizumab (Language: Chinese). *Chinese Clinical Oncology*. 2020,25(9):840-848. DOI:10.3969/j.issn.1009-0460.2020.09.012.
4. Huang J, Xu J, Chen Y, et al. Camrelizumab versus investigator's choice of chemotherapy as second-line therapy for advanced or metastatic oesophageal squamous cell carcinoma (ESCORT): a multicentre, randomised, open-label, phase 3 study. *The Lancet Oncology* 2020; 21:832-42.
5. Wang F, Qin S, Sun X, et al. Reactive cutaneous capillary endothelial proliferation in advanced hepatocellular carcinoma patients treated with camrelizumab: data derived from a multicenter phase 2 trial. *Journal of hematology & oncology* 2020; 13:47.

Comment 9. *Since the IC backbone is not standard, and there are two new agents added at the same time, what should plans be for discerning contribution of each new component of this treatment to the promising outcome?*

Response: We appreciate reviewer's expertise input. Indeed, our recent published phase 3 randomized clinical trial has shown that induction chemotherapy with the regimen of TPC (paclitaxel, cisplatin and capecitabine) is more efficacious than that with PF (cisplatin and CI 5-FU) for patients with stage IVA to IVB NPC (*Li WZ, et al. JAMA Oncology 2022*). Currently, the TPC regimen has become Level I recommendation with Class IA evidence in the Chinese Society of Clinical Oncology (CSCO) guideline. However, none of the N3 patients who received TPC regimen reached CR status, and 2-year PFS rate with TPC regimen was only 85%. Therefore, we assume that the addition of the 2 new biological drugs likely improve the therapeutic efficacy of standard of care treatment. Meanwhile, based on our previous trial and clinical experience of paclitaxel, the incidence of its allergy was 9% (*Li WZ, et al. JAMA Oncology 2022*), while no allergic event occurred in NAB-paclitaxel (*Ke LR, et al. Oral oncology 2017*). Additionally, our recent phase 2 trial has shown encouraging anti-tumor effects and manageable toxicities in locally advanced NPC using NAB-paclitaxel (*Ke LR, et al. Oral oncology 2017*). In order to decrease dose-limiting toxicities of paclitaxel but improve antitumoral efficacy through increasing intratumoral concentration of paclitaxel, we replaced the paclitaxel with NAB-paclitaxel in the present trial. It is worth to mention that we are finishing a clinical trial initiated in 2019 to compare the efficacy of NAB-TPC versus GP in advanced NPC and the upcoming public data shows that NAB-TPC regimen as a first-line treatment provided superior PFS versus GP regimen for advanced NPC (Median IRC-assessed PFS, 11.3 m versus 7.7 m, p=0.002; chicttr.org.cn registration number: ChiCTR1900027112). It is impressive that the QUINTUPLED regimen obtained a high CR rate of 26% after induction phase. We agree with that it is hard to tease out the real factors that contribute to this higher response rate given the nature of the trial design. However, in our previously reported

phase 3 clinical trial (Li WZ, et al. *JAMA Oncology* 2022), we observed a much lower but similar CR rate to IC between TPC and PF regimen (1.7% and 3.3%, respectively), which indicates that the higher CR rate would derive from the addition of the two new drugs rather than the switch from CI 5FU to capecitabine. Meanwhile, in terms of the switch from CI 5FU to capecitabine, similar tumor response rates were observed between 5-FU and capecitabine groups in both anal cancer and advanced metastatic gastric cancer (Jones CM, et al, *Int J Radiat Oncol Biol Phys* 2018; Chen, J et al, *Asia Pac J Clin Oncol* 2018), as well as in rectal cancer (O'Connell MJ, et al, *J Clin Oncol* 2014). Therefore, the addition of the 2 new drugs would be the main contributor for the higher response rate. Indeed, based on the impressive results in the present phase 2 trial, we have designed phase 3 multicenter, randomized clinical trial (QUINTUPLED versus TPC regimen) and will start recruitment soon.

Reference:

1. Li WZ, Lv X, Hu D, et al. Effect of Induction Chemotherapy with Paclitaxel, Cisplatin, and Capecitabine vs Cisplatin and Fluorouracil on Failure-Free Survival for Patients With Stage IVA to IVB Nasopharyngeal Carcinoma: A Multicenter Phase 3 Randomized Clinical Trial. *JAMA oncology* 2022; 8:706-14.
2. Ke LR, Xia WX, Qiu WZ, et al. A phase II trial of induction NAB-paclitaxel and cisplatin followed by concurrent chemoradiotherapy in patients with locally advanced nasopharyngeal carcinoma. *Oral oncology* 2017; 70:7-13.
3. Jones CM, Adams R, Downing A, et al. Toxicity, Tolerability, and Compliance of Concurrent Capecitabine or 5-Fluorouracil in Radical Management of Anal Cancer with Single-dose Mitomycin-C and Intensity Modulated Radiation Therapy: Evaluation of a National Cohort. *International journal of radiation oncology, biology, physics* 2018;101:1202-11.
4. Chen J, Xiong J, Wang J, Zheng L, Gao Y, Guan Z. Capecitabine/cisplatin versus 5-fluorouracil/cisplatin in Chinese patients with advanced and metastatic gastric cancer: Re-analysis of efficacy and safety data from the ML17032 phase III clinical trial. *Asia-Pacific journal of clinical oncology* 2018; 14: e310-e6.
5. O'Connell MJ, Colangelo LH, Beart RW, et al. Capecitabine and oxaliplatin in the preoperative multimodality treatment of rectal cancer: surgical end points from National Surgical Adjuvant Breast and Bowel Project trial R-04. *Journal of clinical oncology: official journal of the American Society of Clinical Oncology* 2014; 32:1927-34.

Comment 10. *I would discourage any graphs showing survival data of non – randomized non-concurrently rerated patients. This over represents the meaningfulness of the comparison.*

Response: Thanks for your suggestion. We have revised Figure 2, removing the survival curve of historical cohort from Figure 2 accordingly.

Comment 11. *It is unclear from the attached protocol how tumor volumes and LN volumes for radiation were determined. While not a primary endpoint issue, with all of these chemoradiation studies going on in NPC, it is essential for the community to understand how RT volumes, when tumor and LN volumes are changing with IC, are being addressed. Are all plans based upon initial pre IC staging and not modified at all based upon responses to IC? Or are plans based on some*

modification based upon IC response?

Response: Thanks for your comments and suggestions. All patients were treated with IMRT definitively. Target volumes were defined in accordance with the International Commission on Radiation Units and Measurements (ICRU) reports 50 and 62. Within 1-2 weeks after the third course of induction chemotherapy, all patients were immobilized in the supine position and using a thermoplastic mask that covered the head, neck, and shoulder. Both non-enhanced CT (for dose calculation) and contrast-enhanced CT (for target delineation) images were obtained from the vertex to 2 cm below the sternoclavicular joint, with 3-mm slices. And MRI fusion with simulation CT images were performed to assist the targets delineation. The primary tumor volumes were defined as the volume sum of all the known gross disease of soft tissues after IC and violated bone before treatment. The LN volumes was defined as the volume sum of all the known gross disease after IC. The high-risk clinical target volume ($CTV_{\text{high-risk}}$) was defined as the volume of a subclinical disease consisting of a 5-10 mm margin surrounding the post-IC GTVnx (2-3 mm posteriorly if adjacent to the brainstem or spinal cord) to encompass the high-risk sites of microscopic extension and the whole nasopharynx. The low-risk clinical target volume ($CTV_{\text{low-risk}}$) was defined as the high-risk clinical target volume plus a 5-10 mm margin (2-3 mm posteriorly if adjacent to the brainstem or spinal cord) to encompass the low-risk sites of microscopic extension, including the skull base, clivus, sphenoid sinus, parapharyngeal space, pterygoid fossa, retropharyngeal nodal regions, and elective neck area from level IB to level V. In order to avoid the misunderstanding, we have specifically described that element in the **Supplement Appendix** of revised manuscript (In the section of Description of the guidelines for intensity-modulated radiation therapy in this trial).

Responses to Reviewer #3

Comment 1. *In the study, the investigators used NAB-TPC as induction chemotherapy regimen rather than gemcitabine and cisplatin, which is the standard of care for locoregionally advanced nasopharyngeal carcinoma. The authors should explain why. Is there any evidence supports that NAB-TPC is more efficacious than gemcitabine and cisplatin as induction chemotherapy?*

Response: We appreciate reviewer's expert input. Indeed, our recent published phase 3 randomized clinical trial has shown that induction chemotherapy with the regimen of TPC (paclitaxel, cisplatin and capecitabine) is more efficacious than that with PF (cisplatin and CI 5-FU) for patients with stage IVA to IVB NPC (*Li WZ, et al. JAMA Oncology 2022*). Currently, the TPC regimen has become a Level I recommendation with Class IA evidence in the Chinese Society of Clinical Oncology (CSCO) guideline. Meanwhile, based on our previous trial and clinical experience of paclitaxel, the incidence of its allergy was 9%, while no allergic event occurred in NAB-paclitaxel. Nevertheless, our recent phase 2 trial has shown NAB-paclitaxel has encouraging antitumoral effects and manageable toxicities in patients with locally advanced NPC (*Ke LR, et al. Oral oncology 2017*). In order to decrease dose-limiting toxicities of paclitaxel but improve anti-tumor efficacy via increasing intratumoral concentration of paclitaxel, we replace the paclitaxel with NAB-paclitaxel in the present trial. It is worth to mention that we are finishing a clinical trial initiated in 2019 to compare the efficacy of NAB-TPC versus GP in advanced NPC. The upcoming public data shows that NAB-TPC regimen as a first-line treatment provided superior PFS versus GP regimen for advanced NPC (Median IRC-assessed PFS, 11.3 m versus 7.7 m, p=0.002; chictr.org.cn registration number: ChiCTR1900027112).

Reference:

1. Li WZ, Lv X, Hu D, et al. Effect of Induction Chemotherapy with Paclitaxel, Cisplatin, and Capecitabine vs Cisplatin and Fluorouracil on Failure-Free Survival for Patients with Stage IVA to IVB Nasopharyngeal Carcinoma: A Multicenter Phase 3 Randomized Clinical Trial. *JAMA oncology* 2022; 8:706-14.
2. Ke LR, Xia WX, Qiu WZ, et al. A phase II trial of induction NAB-paclitaxel and cisplatin followed by concurrent chemoradiotherapy in patients with locally advanced nasopharyngeal carcinoma. *Oral oncology* 2017; 70:7-13.

Comment 2. *In a previous phase II study evaluating camrelizumab plus apatinib in patients with recurrent or metastatic nasopharyngeal carcinoma, investigators reported that the most common complication was nasopharyngeal necrosis (9/16; 56.3%) (J Clin Oncol. 2023 May 10;41(14):2571-2582). How many patients in the present study have nasopharyngeal necrosis? The schedule of apatinib has been modified in second version of protocol given safety concern in the present study. The safety concern should be clarified.*

Response: Thanks for your critical comments. In our study, patients received apatinib only in induction phase and stopped apatinib one week before receiving radiation with the consideration of potential nasopharyngeal necrosis. Indeed, no nasopharyngeal necrosis events have been reported among NPC patients who have not received radiotherapy. Interestingly, we did not observe the nasopharyngeal necrosis in patients treated in this trial.

The usage of apatinib has been modified during the study. In the initial stages of enrollment, we found that many patients were unable to tolerate continuous oral dosing and experienced

discontinuation (detailed information of compliance to apatinib in Supplement Table S5). Thereafter, we amended the protocol with modification of apatinib dose and schedule. The detailed information about the amendment was reported in the Summary of changes in protocol. The compliance data of the apatinib was also reported in **Supplement Table S5**.

Responses to Reviewer #4

Comment 1. *In the background, it's clear that there's a significant issue with distant metastasis in N3 disease patients. However, for greater emphasis and clarity, you might consider reiterating the main problem statement. This will give readers a clearer sense of the problem's importance.*

Response: Thanks for your valuable recommendation. We have revised the statement that “Here, we evaluated the efficacy and safety of the combination of camrelizumab and apatinib with IC and CCRT as a curative approach in stage TanyN3M0 NPC with the goal of significantly decreasing risk of distant metastasis.” in the section of Background in revised manuscript.

Comment 2. *The predetermined values (like DMFS rates of $\leq 76\%$ being considered unacceptable) are specific but might need justification. Why was this particular percentage chosen as the cutoff? This clarity can help readers understand the context better.*

Response: Thanks for your expertise input. At the time of design of this study in 2019, several studies focusing on N3 disease were identified and summarized in the following **Table 1**. There were several common points observed from these studies. Firstly, the majority of N3 patients received high-intensity treatment, including induction/adjuvant therapy and CCRT. Secondly, distant metastasis is the most important determinant of the survival rates for N3 patients despite the availability of aggressive treatment strategies of chemotherapy and radiation therapy. Thirdly, about half of distant metastasis events occurred within 1 year. Fourthly, distant metastasis control rates were similar but unsatisfactory. Given the quality of the literature (decided by NOS, as shown in the NOS column of Table 1) and much more detailed and reliable data derived from cohorts in our cancer center, we eventually decide the sample size based this reference.

Table 1. Summary of trials for N3 NPC patients before design of the present trial.

Year	Area	RT	IC	Cases	No. of N3	DMFS rates	NOS*	Note
2010	Hong Kong ¹	IMRT	Yes	173	15	67.3% (2 y)	8	
2011	Guang Zhou ²	2DRT	Yes	764	232 (N2-3)	65.4% (5 y)	8	
		IMRT	Yes	512	169 (N2-3)	60.9% (5 y)		
2013	Guang Zhou ³	IMRT	Yes	198	26	50.1% (5 y)	7	
2014	Guang Zhou⁴	IMRT	Yes	868	29	62.1% (2 y)	8	76% (1 y)
2017	Fu Zhou ⁵	IMRT	Yes	22	22	81.8% (3 y)	6	WHO II 32%
2017	Nan Jing ⁶	IMRT	Yes	614	40	62.5% (2 y)	7	80% (1 y)
2018	Hang Zhou ⁷	IMRT	Yes	110	110	65% (5 y)	3	Low quality
2019	Fu Zhou ⁸	IMRT	Yes	130	109	70.3% (3 y)	6	~88% (1 y) WHO II 9.2%

*Newcastle-Ottawa Scale score is used for assess the quality, including high quality for 7-9 score, middle quality for 4-6, and lower than 4 for low quality.

Reference:

1. Wong FC, Ng AW, Lee VH, et al. Whole-field simultaneous integrated-boost intensity-modulated radiotherapy for patients with nasopharyngeal carcinoma. International journal of radiation oncology, biology, physics 2010;76:138-45.

2. Lai SZ, Li WF, Chen L, et al. How does intensity-modulated radiotherapy versus conventional two-dimensional radiotherapy influence the treatment results in nasopharyngeal carcinoma patients? *International journal of radiation oncology, biology, physics* 2011;80:661-8.
3. Sun X, Zeng L, Chen C, et al. Comparing treatment outcomes of different chemotherapy sequences during intensity modulated radiotherapy for advanced N-stage nasopharyngeal carcinoma patients. *Radiation oncology (London, England)* 2013;8:265.
4. Sun X, Su S, Chen C, et al. Long-term outcomes of intensity-modulated radiotherapy for 868 patients with nasopharyngeal carcinoma: an analysis of survival and treatment toxicities. *Radiation oncology: journal of the European Society for Therapeutic Radiology and Oncology* 2014; 110: 398-403.
5. Zhang Y, Chen M, Chen C, Kong L, Lu JJ, Xu B. The efficacy and toxicities of intensive induction chemotherapy followed by concurrent chemoradiotherapy in nasopharyngeal carcinoma patients with N3 disease. *Scientific reports* 2017; 7: 3668.
6. Wu LR, Liu YT, Jiang N, et al. Ten-year survival outcomes for patients with nasopharyngeal carcinoma receiving intensity-modulated radiotherapy: An analysis of 614 patients from a single center. *Oral oncology* 2017; 69: 26-32.
7. Chen J, Liu T, Sun Q, Hu F. Clinical and prognostic analyses of 110 patients with N3 nasopharyngeal carcinoma. *Medicine* 2018; 97: e13483.
8. Zong J, Xu H, Chen B, et al. Maintenance chemotherapy using S-1 following definitive chemoradiotherapy in patients with N3 nasopharyngeal carcinoma. *Radiation oncology (London, England)* 2019;14:182.

Comment 3. At line 164, a minor typo exists: it should likely be "StataCorp" instead of "StateCorp."

Response: Thanks for your carefully reviewing. We have corrected it with the word of "StataCorp".

Comment 4. At line 168, "Of them, 1 patient whose diagnosis did not meet eligibility criteria were excluded from the study". It would be more grammatically correct to state "was excluded" instead of "were excluded".

Response: Thanks for your carefully reviewing. We have corrected it using "was".

Comment 5. The interpretation of certain statistics could be made more explicit. For example, while p-values are not mentioned in the results, as they were included in Figure 2, ensuring their interpretation is clear would be beneficial.

Response: Thanks for your comments. Figure 2 has been revised with single curve in revised manuscript.

Comment 6. In the discussion section, the author mentioned that the DMFS rate improvement of 11.7% is presented without context. There should be a discussion on the clinical significance of this improvement.

Response: Thanks for your suggestive comments. We have amended the statement in revised manuscript with the sentence of "In our study, the 2-year rate of DMFS among N3 patients was

significantly improved in comparison to historical cohorts (50-80%) treated with IC and CCRT”.

Comment 7. *As an open-label, single-arm, phase II study, the results might be prone to biases. Was the study population diverse enough to ensure the results were generalizable?*

Response: Thanks for your critical comments. The proportion of NPC patients with N3 disease is relatively small (about 10%) among patients with locoregionally advanced NPC (LA-NPC), but these patients have the highest risk of recurrence and distant metastasis. As reported in *Zhang et al, N Engl J Med 2019*, there are only a total 51 N3 disease enrolled among all 480 LA-NPC patients. The required sample size was calculated based on previous trial data with the statistical calculation. Although we completely agree with that a large, randomized control trial is warranted, our reported results in this manuscript showed that the QUINTUPLED regimen obtains promising distant metastasis control in stage TanyN3M0 NPC. Indeed, based on these impressive results in the present phase 2 trial, we have designed a phase 3 multicenter and randomized clinical trial and will start recruitment soon.

Reference:

1. Zhang Y, Chen L, Hu GQ, et al. Gemcitabine and Cisplatin Induction Chemotherapy in Nasopharyngeal Carcinoma. *The New England journal of medicine* 2019; 381:1124-35.

REVIEWER COMMENTS

Reviewer #1 (Remarks to the Author):

The authors have done a round of extensive edits and responded to our comments. I still have the following suggestions and clarifications that will help enhance the quality of the manuscript.

1. Abstract: It is not clear that camrelizumab was continued during RT and maintained for 12 months total in the Abstract. It seemed like apatinib and camrelizumab was only added in the induction phase. Outlining the regimen in detail in the Abstract will be helpful to the readers.
2. Abstract: Since the rate of relapse is low especially in the first year (DMFS 98%), this reviewer suggests including the results on how many relapse or death events occurred within the first 2 years.
3. Methods (Procedures): This reviewer would request that the authors at least mention the RT doses that were prescribed for the respective CTVs. Additionally, why was level 1b mandatory in the treatment volume since coverage of this lymph node station isn't standard practice; was it done for all cases or only for a subset of patients.
4. Methods (Procedures): This reviewer thanks the authors for clarifying on the follow-up protocol. However, it seems a little excessive to be doing CT staging every 3 months for the first 3 years. Can the authors indeed confirm that was performed? Was EBV DNA used for surveillance since this was included in the CSCO guidelines. If so, it seems sensible to rely on EBV DNA surveillance and a positive result would prompt CT restaging.
5. Results (Table 2): This reviewer is still unconvinced by the lymph node biopsy results. Ultimately, even though this procedure was undertaken at a high volume centre by experts, it is highly unlikely that all the lymph nodes could be biopsied in these patients with bulky nodes. Moreover, this was only performed in a subset of patients, so bias couldn't be entirely excluded. Thus, this reviewer is sceptical about the accuracy of this result.

In fact, I would counter propose that the authors include EBV DNA response in this Main Table, as opposed to placing it in the Supplementary Table 7, since this is a Main result.

6. Finally, this reviewer would like the authors to comment on the next steps going forward. It is unlikely that patients in the real-world can tolerate a 5-drug regimen, especially when apatinib is not an easy drug to tolerate. Would future iterations involve gemcitabine+apatinib+camrelizumab or other anti-PD1 or gemcitabine+cisplatin+apatinib+anti-PD1?

Reviewer #2 (Remarks to the Author):

While there is lengthy discussion between other reviewers and authors in the rebuttal document covering many issues, the fundamental issue at the core of my opinion originally rendered is unchanged. One can only draw so many conclusions in terms of promising outcome from an uncontrolled study where there are several changes in treatment made simultaneously. The observations are interesting, preliminary, come in the context of a small uncontrolled data set. See some specifics below.

1. I remain concerned about overrepresentation of results from this small, uncontrolled trial. Specifically “In our study, the 2-year rate of DMFS among N3 patients was significantly improved in comparison to historical cohorts (50%-80%) treated with IC and CCRT4,9,23,24,25”

A more accurate representation would be” higher “ DMFS rather than “ improved” . While this may seem like hair splitting, it really represents the difference between small uncontrolled dataset and large controlled dataset.

This issue is repeated later as well:” In fact, the present 278 combination regimen significantly improved both FFS and DMFS compared with traditional regimen used in our historical N3 cohort9”

The authors do call for further study but should temper language such as “improved” in this context.

2. The English language usage is still very suboptimal. See lines 49, 53, 267, 272, 289, 313 as examples

3. The comment in the discussion that “ this new strategy would be a promising first-line

option for patients with locally advanced NPC” is possibly misleading. What is meant by “would” ? Are the investigators recommending adoption or definitive testing? Similar issues with “In our study, 85% and 96% patients had 256 undetectable EBV DNA levels after IC and CCRT, respectively, indicating this 257 combination regimen did effectively eradicate micrometastatic lesions and imply 258 satisfactory long-term distant control”.

Reviewer #3 (Remarks to the Author):

The authors have addressed all of my concerns. I have no further questions.

Reviewer #4 (Remarks to the Author):

The authors have adequately addressed the comments made by the reviewers in the revised version of the manuscript. Therefore, I have no further comments.

Responses to Reviewer #1

Comment 1. Abstract: *It is not clear that camrelizumab was continued during RT and maintained for 12 months total in the Abstract. It seemed like apatinib and camrelizumab was only added in the induction phase. Outlining the regimen in detail in the Abstract will be helpful to the readers.*

Response: Thank you for your constructive comments. In order to reduce confusion, we have detailed the regimen as the following in the abstract: 49 eligible patients with stage T_{any}N3M0 NPC were enrolled and received the combination of three cycles of induction chemotherapy (nab-paclitaxel 200 mg/m² on day 1, cisplatin 60 mg/m² on days 1, and capecitabine 1000 mg/m² twice daily on days 1 to 14), camrelizumab and apatinib (250 mg 5 days per week for 8 weeks) followed by chemoradiotherapy (once-every-3-weeks cisplatin 100 mg/m² on days 1 for 2 cycles). Camrelizumab was administered intravenously at a fixed dose of 200 mg every 3 weeks starting from the first cycle of induction chemotherapy until 12 months.

Comment 2. Abstract: *Since the rate of relapse is low especially in the first year (DMFS 98%), this reviewer suggests including the results on how many relapse or death events occurred within the first 2 years.*

Response: Thank you for your recommendation. We have added the information in the abstract as the following: the primary endpoint of 1-year distant metastasis-free survival was 98% (one death due to liver and bone metastases, 95% confidence interval of 88-100). The 2-year failure-free survival was 96% (1 recurrence and 1 death due to metastasis).

Comment 3. Methods (Procedures): *This reviewer would request that the authors at least mention the RT doses that were prescribed for the respective CTVs. Additionally, why was level Ib mandatory in the treatment volume since coverage of this lymph node station isn't standard practice; was it done for all cases or only for a subset of patients.*

Response: Thanks for your comment. We have added the additional information in the section of Procedures, Methods in the revised manuscript as following: For the planning target volumes of GTV_{nx}, GTV_{nd}, CTV_{high-risk}, and CTV_{low-risk}, total radiation doses of 68–72 Gy, 64–68 Gy, 60–64 Gy, and 54–58 Gy, respectively, were administered in 30-33 fractions, delivered daily at five fractions per week.

According to the international guidelines for the delineation of the clinical target volumes for NPC and the CSCO clinical guidelines (*Lee AW et al, Radiotherapy and Oncology 2018; , Tang LL et al, Cancer Commun 2021*), preventive irradiation on area Ib lymph nodes is required only for high-risk patients in the real-world practice, specifically, including patients with submandibular gland involvement, diseases involving the anatomical structure of the lymph node drainage area (oral and nasal front half) with area Ib as the first stop, lymph node invasion in area II accompanied by extracapsular invasion, or in cases where the maximum diameter of the lymph node in area II exceeds 2 cm.

In order to avoid misunderstanding, we have revised the description in the section of Description of the guidelines for intensity-modulated radiation therapy in this trial in Supplementary Appendix as the following: Preventive irradiation on Area Ib will be applied in the following high-risk groups: submandibular gland involvement, disease involving the anatomical structure of the lymph node drainage area (oral and nasal front half) with area Ib as the first stop, lymph node invasion in area II accompanied by extracapsular invasion, or in cases where the maximum diameter

of the lymph node in area II exceeds 2 cm.

Reference:

1. Lee AW, Ng WT, Pan JJ, et al. International guideline for the delineation of the clinical target volumes (CTV) for nasopharyngeal carcinoma. *Radiotherapy and oncology: journal of the European Society for Therapeutic Radiology and Oncology* 2018;126:25-36.
2. Tang LL, Chen YP, Chen CB, et al. The Chinese Society of Clinical Oncology (CSCO) clinical guidelines for the diagnosis and treatment of nasopharyngeal carcinoma. *Cancer Commun (Lond)* 2021;41:1195-227.

Comment 4. Methods (Procedures): This reviewer thanks the authors for clarifying on the follow-up protocol. However, it seems a little excessive to be doing CT staging every 3 months for the first 3 years. Can the authors indeed confirm that was performed? Was EBV DNA used for surveillance since this was included in the CSCO guidelines. If so, it seems sensible to rely on EBV DNA surveillance and a positive result would prompt CT restaging.

Response: Thank you for your comment. As described in the trial protocol, patients were followed up every 3 months during the first 3 years and every 6 months thereafter until death after completion of radiotherapy. The CT examination was done on all patients, but not at every follow-up timepoint, which depended on individual patient's financial burden and the patient's complaints. That being said, we had other alternative measurements to evaluate/monitor disease status, including endoscopy, MRI, chest radiograph and abdominal ultrasound if indicated. Indeed, the EBV DNA surveillance is tested routinely at every follow-up timepoint in our trial. We agree with the comment that the combination of regular EBV DNA surveillance with subsequent CT restaging (with positive EBV) surveillance may be sensible.

Comment 5. Results (Table 2): This reviewer is still unconvinced by the lymph node biopsy results. Ultimately, even though this procedure was undertaken at a high volume centre by experts, it is highly unlikely that all the lymph nodes could be biopsied in these patients with bulky nodes. Moreover, this was only performed in a subset of patients, so bias couldn't be entirely excluded. Thus, this reviewer is skeptical about the accuracy of this result.

In fact, I would counter propose that the authors include EBV DNA response in this Main Table, as opposed to placing it in the Supplementary Table 7, since this is a Main result.

Response: We thank you for this comment. Since the curative treatment modality for NPC is radiotherapy rather than surgery, it is difficult to determine the pathologic response by sampling lymph nodes with biopsy alone. We agree that the lymph node biopsy is unreliable for evaluation response. We only considered the results derived from lymph node biopsy as part of the exploratory research objective. Because of these considerations, we did not use the concept of pCR in the results, but instead used the term “pathological evaluation of biopsy”. In addition, as suggested, we have integrated the content included in original Supplementary Table S7 into Table 2 as shown below.

Table 2. Response to Treatment, Pathological evaluation, and Plasma EBV DNA clearance information.

	no./total no. (%)
Response to induction phase	

Objective response	49/49 (100.0)
Complete response	13/49 (26.5)
Partial response	36/49 (73.5)
Stable disease	0
Progressive disease	0
Endoscopic biopsy	
After 3 rd cycle of induction phase	
Negative	28/33 (84.8)
Positive	5/33 (15.2)
Lymph nodes biopsy	
After 1 st cycle of induction phase	
Negative	9/23 (39.1)
Positive	14/23 (60.9)
After 2 nd or 3 rd cycle of induction phase	
Negative	8/11 (72.7)
Positive	3/11 (27.3)
Response to radiation phase*	
Objective response	46/47 (97.9)
Complete response	45/47 (95.7)
Partial response	1/47 (2.1)
Stable disease	0
Disease recurrence	1/47 (2.1)
Plasma EBV DNA clearance [†]	
Persistently detectable	2/48 (4%)
Change to undetectable	46/48 (96%)
Median (IQR) clearance time, days	36/48 (27-56)
Range, days	7-106
Undetectable during induction therapy phase	41/48 (85%)
After 1 st cycle of induction therapy	15/48 (31%)
After 2 nd cycle of induction therapy	17/48 (35%)
After 3 rd cycle of induction therapy	9/48 (19%)
Undetectable during radiotherapy phase	5/48 (10%)

IQR = interquartile range.

* Two patients were excluded, including one patient who declined radiotherapy after induction phase and the other patient whose radiotherapy was interrupted prematurely after receiving 9 fractions due to personal reasons. Both of them remained in complete response at the time of the latest follow-up.

† There are 48 patients who had positive plasma Epstein-Barr virus (EBV) DNA level at baseline. Quantification testing of plasma EBV DNA level was based on a lower limit of detection cutoff value of 1000 copies/mL. The undetectable levels refer to individuals whose EBV DNA levels are reported at 0 copies/mL.

Comment 6. *Finally, this reviewer would like the authors to comment on the next steps going forward. It is unlikely that patients in the real-world can tolerate a 5-drug regimen, especially*

when apatinib is not an easy drug to tolerate. Would future iterations involve gemcitabine+apatinib+camrelizumab or other anti-PD1 or gemcitabine+cisplatin+apatinib+anti-PD1?

Response: Thank you for your expert and helpful suggestion. The combination regimen did not result in any unexpected grade 3 or 4 TRAEs or irAEs in the trial. The cumulative TRAE rate of 65% is similar to other reported studies (ranging from 58% to 76%). Previous studies showed that the proportion of patients completing 3 cycles of IC and 2 cycles of CCRT ranges from 88% to 97% and 88% to 92%, respectively. In the present study, 100% (49/49) and 87% (41/47) of the patients received and finished 3 cycles of induction chemotherapy and 2 cycles of CCRT, respectively. This suggests that the additional camrelizumab and apatinib to induction chemotherapy does not increase TRAE rate and the combination regimen is well tolerated.

During the trial, the dosage of apatinib was amended based on the safety concern raised from the first portion of enrolled patients. At the beginning of the study, the schedule and dosage of apatinib was designed as to start 250 mg apatinib daily from day 1 to 56. With this dosage, we found that more than half of patients (17/25, 68%) were unable to tolerate continuous oral dosing and experienced discontinuation (detailed information of compliance to apatinib is shown in Supplementary Table S5). Thereafter, we modified the dosage of apatinib in the amended protocol. The change was reported in the Summary of changes in the protocol. As a result, more patients tolerated the treatment as compared to original dosing (p value = 0.032), and only 9/24 (37.5%) patients experienced dose interruptions (The compliance data of the apatinib was reported in Supplementary Table S5).

Our anticipated next step is to launch a randomized Phase 3 clinical trial to compare the present QUINTUPLED regimen to gemcitabine + cisplatin + camrelizumab regimen in NPC patients with N3 disease and further validate the findings. Indeed, it is worth to mention that we are finishing a clinical trial initiated in 2019 to compare the efficacy of NAB-TPC versus GP regimen in advanced NPC. The upcoming public data shows that NAB-TPC regimen as a first-line treatment provided superior PFS versus GP regimen for advanced NPC (Median IRC-assessed PFS, 11.3 m versus 7.7 m, p=0.002; chictr.org.cn registration number: ChiCTR1900027112). Therefore, based on our previous studies (*Li WZ, et al. JAMA Oncology 2022; study of ChiCTR1900027112. the upcoming public data*), the NAB-TPC regimen will be the preferred chemotherapy regimen backbone for our subsequent studies.

Reference:

1. Li WZ, Lv X, Hu D, et al. Effect of Induction Chemotherapy With Paclitaxel, Cisplatin, and Capecitabine vs Cisplatin and Fluorouracil on Failure-Free Survival for Patients With Stage IVA to IVB Nasopharyngeal Carcinoma: A Multicenter Phase 3 Randomized Clinical Trial. *JAMA oncology* 2022;8:706-14.

Responses to Reviewer #2

Comment 1. *I remain concerned about overrepresentation of results from this small, uncontrolled trial. Specifically “In our study, the 2-year rate of DMFS among N3 patients was significantly improved in comparison to historical cohorts (50%-80%) treated with IC and CCRT4,9,23,24,25”*

A more accurate representation would be” higher “ DMFS rather than “improved” . While this

may seem like hair splitting, it really represents the difference between small uncontrolled dataset and large controlled dataset.

This issue is repeated later as well:” In fact, the present 278 combination regimen significantly improved both FFS and DMFS compared with traditional regimen used in our historical N3 cohort9”

The authors do call for further study but should temper language such as “improved” in this context.

Response: Thanks for your recommendation. We have revised accordingly in the revised manuscript as suggested on Pages 13 and 14.

Comment 2. *The English language usage is still very suboptimal. See lines 49, 53, 267, 272, 289, 313 as examples.*

Response: Thanks for your comment. We have gone over the whole manuscript with the assistance from a native English speaker and made revision, accordingly, including abovementioned lines 49, 53, 267, 272, 289, 313.

Comment 3. *The comment in the discussion that “this new strategy would be a promising first-line option for patients with locally advanced NPC” is possibly misleading. What is meant by “would”? Are the investigators recommending adoption or definitive testing? Similar issues with “In our study, 85% and 96% patients had undetectable EBV DNA levels after IC and CCRT, respectively, indicating this combination regimen did effectively eradicate micrometastatic lesions and imply satisfactory long-term distant control”.*

Response: Thanks for your comment. We have revised the statements as the following: “this new strategy is a promising first-line option for patients with locally advanced NPC and high risk of distant metastasis after validation in the large-scale phase 3 trial” and “In our study, 85% and 96% patients had undetectable EBV DNA levels after IC and CCRT, respectively, indicating this combination regimen did effectively control EBV titer, which is strongly associated with micrometastatic lesions and disease progression.

REVIEWERS' COMMENTS

Reviewer #1 (Remarks to the Author):

I thank the authors for taking our comments onboard.

I have no further comments on this work.